# DISTRIBUTION-GUIDED LOCAL EXPLANATION FOR BLACK-BOX CLASSIFIERS

## ABSTRACT

Existing local explanation methods provide an explanation for each decision of black-box classifiers, in the form of relevance scores of features according to their contributions. To obtain satisfying explainability, many methods introduce ad hoc constraints into the classification loss to regularize these relevance scores. In this paper, we discuss some shortcomings of these methods and address them by a simple but effective mask predictor. Specifically, we present the concept of distribution controllers and integrate it with a neural network to directly guide the distribution of relevance scores. Then we introduce the classification loss to optimize the proposed predictor. The benefit of this strategy is to enable discriminative scores over supporting features, and facilitate the setting of involved hyperparameters. The experimental results demonstrate that the proposed method also outperforms others in terms of faithfulness and explainability. The code is available at https://github.com/iclrlocal/.

## 1 INTRODUCTION

Deep neural networks (DNNs) have achieved high classification accuracy in a wide range of fields, such as computer vision (He et al., 2016; Simonyan & Zisserman, 2014) and natural language processing (Greff et al., 2016; Mikolov et al., 2010). Despite the superior performance, DNN models lack meaningful explanations on how a specific decision is made, and are often regarded as blackbox classifiers. To address this issue, various global and local explanation methods have been proposed. The former group aims to inspect the structures and the parameters of a complex model (Erhan et al., 2009; Chen et al., 2016). The latter group provides users understandable rationale for a specific decision with relevance scores[1] (Simonyan et al., 2014; Du et al., 2018a).

In this paper, we focus on local explanation as it extracts the intuitive evidence behind the decision of each instance. To obtain the relevance scores for local explanation, gradient-based methods compute the partial derivative of the class probability with respect to an input instance. However, instead of directly pointing out why the target class is derived based on input, it is likely to answer the question (Montavon et al., 2018): What makes this instance more or less similar to the target class? To tackle this limitation, perturbation-based explanation methods are proposed. These methods perturb the input and aim to find the smallest region, which alone allows a confident classification or prevents a confident classification once being removed (Dabkowski & Gal, 2017; Fong & Vedaldi, 2017). By applying various ad hoc constraints, these methods improve explainability and maintains faithfulness[2]. Nevertheless, most of these methods either focus on distinguishing the supporting features from background and ignore the discrimination on these features, or are only able to predict discriminative scores on a small part of supporting features. To understand the behaviour of a classifier, the discriminative scores over all supporting features are preferred. The comparison of an expected salinecy map to others are shown in Fig.1(a)[3]. Moreover, the information gap between the classification loss and the constraints in turn increases the difficulty of tuning hyper-parameters.

---

[1]In this paper, relevance scores indicate the contributions of features to a specific decision. A high score implies a higher contribution. Besides, we do not discriminate saliency maps and masks, as both indicate the permutation of relevance scores of each input.

[2]Explainability quantifies how easy it is to understand and reason about the explanation; faithfulness estimates the fidelity between the explanation and the decision behaviour of black-box classifiers.

[3]Besides highlighting the pixels within the dog for the images of dogs, the relevance scores should also be discriminative at various parts of the dog, such as higher scores on the head than legs.

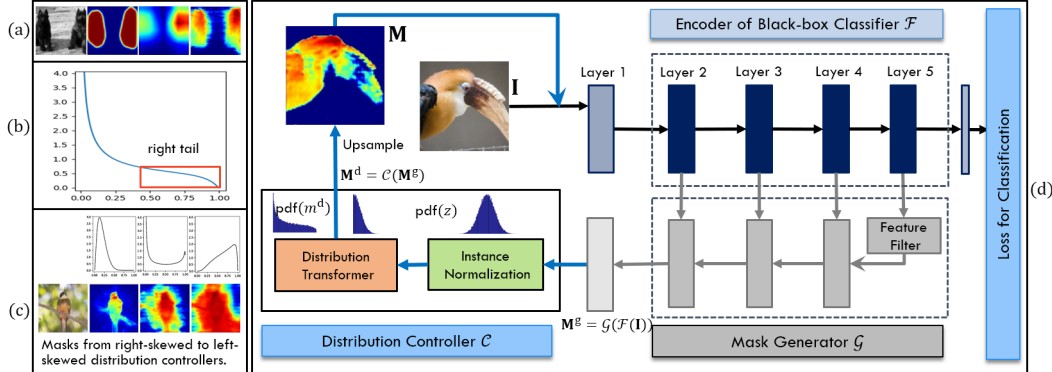

Figure 1: (a) An example of the comparison of the expected saliency map (the last one) to others. (b) An example of right-skewed distributions. (c) The illustration of the benefits of using right-skewed distributions for explanation. (d) The framework of our mask predictor, where a distribution controller is introduced right after the mask generator. In particular, pdf stands for the probability density function.

To address these issue, we propose a simple but effective mask predictor. The work is built upon the following observation: *a large portion of contributions to each decision are held by only a small fraction of features* (Fong & Vedaldi, 2017; Chattopadhay et al., 2018; Du et al., 2018a). The proposed predictor consists of a mask generator and a distribution controller. The former takes the hidden feature maps in black-box classifiers as inputs, and the latter guides the outputs with expected distributions as relevance scores. We show that, with an easy setting of the involved hyper-parameters in the controller, it can directly enforce the relevance scores towards right-skewed distributions (Clauset et al., 2009). Then, the scores of supporting features becomes more discriminative, which correspond to the small portion of the right tail, and the majority features have low scores and are regarded as unimportant. For illustration, an example of a right-skewed distribution is shown in Fig.1(b), and the masks predicted based on the distributions controllers with the right-skewed to the left-skewed are displayed in Fig.1(c). Furthermore, we introduce classification losses to optimize mask predictors, which removes all constraints and avoid non-trivial hyper-parameter tuning. We show that such a simplifications will not sacrifice the explainability too much with an ablation study. Finally, we introduce two metrics for comprehensively evaluating relevance scores in terms of faithfulness and explainability, respectively. The main contributions of our work are as follows.

- We present the conception of distribution controllers on relevance scores, and integrate it with a trainable mask generator to improve the discrimination of supporting features. We provide two practical implementations of controllers to enforce scores towards the desired right-skewed distributions, where the involved hyper-parameters can be easily set.
- We introduce the classification loss to train the proposed model. It avoids the non-trivial hyper-parameter tuning on ad hoc constraints and also improves the faithfulness of mimicking target black-box classifiers.
- We empirically demonstrate the effectiveness of the above innovations. Specifically, we change the setting of distributions and show the controllers can guide the scores towards varying preset distributions. Besides, we perform an ablation study by adding constraints back to analyze their effect on masks. Furthermore, the experiments also demonstrate that our method not only obtains higher quantitative performance for explaining the behavior of black-box classifiers, but also provides discriminative masks for intuitive explanation.

## 2 RELATED WORK

This section reviews local explanation methods for DNNs, which target to identify the relevance score of each feature towards a specific decision (Montavon et al., 2018).

### 2.1 GRADIENT-BASED LOCAL EXPLANATION

To obtain relevance scores, gradient-based methods compute the partial derivative of the class probability with respect to the input by using back-propagation (Simonyan et al., 2014). In general, these

methods are advantageous in their high computational efficiency, i.e., using a few forward and backward iterations is sufficient to generate saliency maps. However, these saliency maps based on the naive gradients are visually noisy and hard to be understand. To address this issue, various methods have been proposed. For example, Smooth Grad (Smilkov et al., 2017) addresses the visual noise by introducing noise to inputs repeatedly. Integrated gradient (Fong & Vedaldi, 2017) estimates the global contribution of each feature rather than the local sensitivity. Guided back-prorogation (Springenberg et al., 2014) modifies the gradients of RELU functions by discarding negative values at the back-propagation process. Besides, recent methods proposed to create saliency maps by combining the gradients with the corresponding features. For example, Grad CAM (Selvaraju et al., 2017) and Grad CAM ++ (Chattopadhay et al., 2018) take advantages of high-level feature maps, which makes their saliency maps more clear. Nevertheless, since black-box classifiers are trained without any location information, the object locations in their high-level layers may not always correspond to the locations in raw images, leading to new issues for these explanation methods.

## 2.2 PERTURBATION-BASED LOCAL EXPLANATION

The perturbation-based methods first perturb an input according to a given mask, and then observe the new class probability of black-box classifiers. By measuring its difference to the probability of the raw input, the supporting features in the input with the class label can be located (Dabkowski & Gal, 2017). Let $\mathbf{M}$ indicate the expected mask of an image $\mathbf{I}$, where $m_{ij} \in \mathbf{M}$ indicates the relevance score of the pixel within $\mathbf{I}$. Besides perturbing $\mathbf{I}$ with mask $\mathbf{M}$, these methods also introduce an alternative background image $\mathbf{A}$ to reduce the amount of unwanted evidences:

$$\phi(\mathbf{I}, \mathbf{M}) = \mathbf{I} \odot \mathbf{M} + \mathbf{A} \odot (\mathbf{1} - \mathbf{M}). \tag{1}$$

Suppose $f_c$ indicates the probability of the predicted class $c$ based on the black-box classifier. These method input the perturbed images to the classifier and optimize the mask with ad hoc constraints:

$$\text{argmin}_{\mathbf{M}} \ell_{\text{Ex}}(\mathbf{M}) = \text{argmin}_{\mathbf{M}} \lambda_{\text{av}} \text{AV}(\mathbf{M}) + \lambda_{\text{tv}} \text{TV}(\mathbf{M}) - f_c(\phi(\mathbf{I}, \mathbf{M})) + \lambda_{\text{bg}} f_c(\phi(\mathbf{I}, \mathbf{1} - \mathbf{M}))$$
$$\tag{2}$$

where AV aims to minimize the average of all scores and TV enforces the mask to be smooth. The last two terms aim to obtain discriminative scores between supporting pixels and the rest pixels.

To improve the explainability, various novel methods following the above formulation are subsequently proposed. For example, (Du et al., 2018b) regularizes the expected mask with middle-level features and optimizes the mask by reconstructing higher-level feature maps. (Fong & Vedaldi, 2017) introduces a deletion game and reformulates the problem by applying multiple masks stochastically. (Fong et al., 2019) introduces extreme perturbation with an area constraint. However, these methods need to optimize the mask for each image individually, leading to non-negligible time costs.

## 3 THE PROPOSED METHOD

In this section, we introduce the details of the proposed framework, as shown in Fig.1(b). It consists of an encoder inside the black-box classifier $\mathcal{F}$, a mask generator $\mathcal{G}$, and a distribution controller $\mathcal{C}$. In particular, the generator and controller together compose the mask predictor, which takes the feature maps of the black-box classifier as inputs and predicts the relevance scores for each instance. In the following sections, we first review the mask generator and then introduce the proposed predictor.

### 3.1 THE REVIEW OF MASK GENERATOR

We first describe the mask generator $\mathcal{G}$ in our framework. Specifically, it contains three bottleneck blocks and follows the U-Net architecture with the black-box classifier (Dabkowski & Gal, 2017). Each image is input into the classifier for producing feature maps at multiple layers. Then the mask generator upsamples the feature maps of lower resolution using transposed convolutions, and concatenates them with the higher-resolution feature maps. Based on multiple transposed convolution layers, the generator produces a reduced mask $\mathbf{M}^{\text{g}} = (\mathcal{G}(\mathcal{F}(\mathbf{I}))$ at a coarse scale and obtain the relevance score on each location. Then, upsampling based on bilinear interpolation is employed to obtain more smooth masks at the image scale. Let $\mathbf{M}$ indicate the expected mask. Denote $\mathcal{F}(\mathbf{I})$ as the feature maps for the image $\mathbf{I}$. The image is then perturbed with the generated mask with Eq.1,

Figure 2: The examples of the ablation studies on (a) (Dabkowski & Gal, 2017) and (b) the proposed method. (c) The negative effect of non-monotonic transformers.

and the generator can be optimized based on the constraints in Eq.2:

$$\arg\min_{\mathcal{G}} \ell_{\mathrm{Ex}}(\mathbf{M}), \text{ where } \mathbf{M} = \mathrm{upsample}(\mathbf{M}^{\mathrm{g}}). \tag{3}$$

In particular, a simple feature filter is pretrained to perform initial localization with respect to the predicted class. Since it is not the focus of this paper, we leave the detailed description in A.1.

The generator module can produce explanation masks in real time. However, the problem of this method is that, it only tries to distinguish the supporting features from others, without considering the discrimination of scores among these supporting features for better explainability. Besides, balancing the trade-offs between the classification loss and additional constraints, such as the smoothing term in Eq. 2), involves a non-trivial hyper-parameter tuning process.

For illustration, an example is shown in Fig.2(a), where the masks are obtained from the trained generators under the following hyper-parameter settings: (1) carefully-tuned $\lambda$s, (2) $\lambda_{\mathrm{av}}$=0, (3) $\lambda_{\mathrm{bg}}$=0, (4) $\lambda_{\mathrm{tv}}$=0, (5) $\lambda_{\mathrm{bg}}$=$\lambda_{\mathrm{tv}}$=0. As we observed, each term in Eq. 2 brings a significant effect on final masks. Of note, although $\lambda_{\mathrm{bg}}$=0 can release the masks to obtain discriminative scores on a small part of supporting pixels, it reduces the explainability on the others. Besides, it increases the sensitiveness of $\lambda_{\mathrm{av}}$ on masks. The reason is that by enforcing $f_c(\phi(\mathbf{I}, \mathbf{1}\text{-}\mathbf{M}))$ to low values, masks can highlight the majority of supporting pixels even when $\lambda_{\mathrm{av}}$ is slightly larger. Once it is removed with $\lambda_{\mathrm{bg}}$=0, the size of high scores is totally controlled by $\lambda_{\mathrm{av}}$. More details are discussed in Sec. 4.4.

### 3.2 THE MASK PREDICTOR

To address the above issues, we first present a distribution controller after the above mask generator to guide the relevance scores. Then, we introduce classification losses for optimizing the predictor without ad hoc constraints. The details of two technical innovations are presented below.

#### 3.2.1 THE CONTROLLER ON DISTRIBUTION

To produce discriminative scores on supporting features in $\mathbf{M}$, we introduce the conception of distribution controllers $\mathcal{C}$, which is used to guide the relevance scores towards desired distributions. Suppose $\mathbf{M}^{\mathrm{g}}$ is the output of the generator on $\mathbf{I}$, and $\mathbf{M}^{\mathrm{d}}$ denotes the output of $\mathcal{C}$. The output of distribution controller is expressed as

$$\mathbf{M}^{\mathrm{d}} = \mathcal{C}(\mathbf{M}^{\mathrm{g}}). \tag{4}$$

Based on the above transform, we expect to encode the distribution inside the mask predictor rather than bringing the formulation an additional constraint. Besides, we follow (Dabkowski & Gal, 2017; Du et al., 2018b) to upsample $\mathbf{M}^{\mathrm{d}}$ with interpolation, aiming to improve the smoothness of masks.

**The principles of controllers.** Now we investigate two principles for designing the controller $\mathcal{C}$.

*Principle 1.* We expect that the output scores of $\mathcal{C}$ approach a right-skewed distribution for each instance. The motivation behind is that a large portion of contributions to each decision are supposed to be held only by a small fraction of supporting features with varying importance. These supporting features, whose information should be preserved by the mask, are assigned with higher but discriminative scores.

*Principle 2.* We impose the monotonic mapping from the distribution controller's input $\mathbf{M}^{\mathrm{g}}$ to its output $\mathbf{M}^{\mathrm{d}}$. The purpose is to enhance the discriminative ability of the coarse mask $\mathbf{M}^{\mathrm{g}}$ but without changing its ranking of relevance scores.

For better illustration, two examples are shown in Fig.1(a) and Fig.2(c). In the first example, with the expected distributions of controllers varying from the right-skewed to the left-skewed, the dis-

crimination of supporting features is remarkably reduced. Besides, the proportion of features that are highlighted with high scores is nearly positive correlation to the area at the right part of preset distributions. In the second example, a non-monotonic transform $g(x) = x^2$ is used in controllers, which results the scores in the deviance from expected meanings. That is, a higher score implies a larger contribution.

**The implementation of controllers.** Following the above principles, we introduce two piratical implementation of the controllers. Since a sum of even two independent random variables is more Gaussian than the original variables (Hyvärinen & Oja, 2000; Ioffe & Szegedy, 2015), we first assume that the input scores of controllers (the outputs of generators) are nearly Gaussian for convenience. Later we show that the proposed controller built upon this assumption can also benefit other distributions towards the right-skewed.

*From the Gaussian to the normal.* In order to obtain the desired distributions, we first introduce instance normalization (Ulyanov et al., 2016) to transform Gaussian distributions towards normal distributions. The goal of this step is to shift scores around two sides of 0.

Specifically, let $m_{ij}^{\mathrm{g}} \in \mathbf{M}^{\mathrm{g}}$ indicates the input entry at the location $(i, j)$ of $\mathcal{C}$, and $z_{ij} \in \mathbf{Z}$ denotes the expected variable with a normal distribution. The mapping can be expressed as

$$z_{ij} = (m_{ij}^{\mathrm{g}} - \mathrm{E}[m_{ij}^{\mathrm{g}}])/(\sqrt{\mathrm{Var}[m_{ij}^{\mathrm{g}}]}), \tag{5}$$

where the expectation $\mathrm{E}[\cdot]$ and variance $\mathrm{Var}[\cdot]$ are computed over the entries of each $\mathbf{M}^{\mathrm{g}}$.

*From the normal to the right-skewed.* Now we further transform the above scores towards a right-skewed distribution monotonically. To do this, we introduce two simple transformers, including a basic one without any hyper-parameter, and a customized one with easy-to-set hyper-parameters.

Basic Transformer (BT): The straightforward way of transforming a normal distribution into a right-skewed distributions is cropping variables with Rectified Linear Unit (ReLU) (Nair & Hinton, 2010). Specifically, the expected output of a distribution controller $m_{ij}^{\mathrm{d}}$ can be obtained as

$$m_{ij}^{\mathrm{g}} = \mathrm{ReLU}(z_{ij}) = \max(0, z_{ij}). \tag{6}$$

The benefit of BT is to enable a monotonic transformation without any hyper-parameter. However, BT is lack of the potential ability to handle other distributions, such as the left-skewed inputs.

Customized Transformer (CT): To address the issues of BT, we introduce an alternative with easy-to-set hyper-parameters. The goal is to produce scores for all features with a right tail in (0,1). Specifically, we first transform the normal distribution towards an uniform distribution based on sigmoid functions, and then change the skewness of the distribution based on power functions:

$$m_{ij}^{\mathrm{d}} = (\mathrm{sigmoid}(\eta \cdot z_{ij}))^h = \left(\frac{1}{1 + \exp^{-\eta \cdot z_{ij}}}\right)^h, \tag{7}$$

where $\eta$ is used to approach the uniform distribution (Waissi & Rossin, 1996), and $h$ determines the skewness of the obtain distribution. Based on the probability density transformation (Forbes et al., 2011), we can easily set these hyper-parameters based on the intuitive geometric properties of their probability density functions. For example, with $\eta$=1.5 and $h$=2.5, we can obtain the scores with the probability density function in Fig.1(b). Owing to the space limitation, we leave the detailed discussion to A.2. Note that other transformers with monotonicity can also be considered.

*The affect on other distributions.* Next, we release the Gaussian assumption and consider the effect of the preset controller on other distributions. Specifically, we introduce the CT with $\eta$=1.5 and $h$=2.5 on these distributions. The original (in gray) and transformed distributions (in blue) are displayed in Figs.3(a-d). As we can see, although CT may not transform them to the right-skewed perfectly, it can still shift a part of scores to low values and make scores away from the left-skewed.

**The post hoc tuning of distributions.** In practice, constraining different images with a same distribution may lead to redundant scores on unimportant pixels, especially when the supporting pixels only take a small part of the image. For this, we present a simple trick to perform a post hoc tuning.

Given a predicted mask, which ranks all image pixels with respect to their scores for the predicted class. We first iteratively add ranked pixels from each image and measure the resulting probability $q_i$ for $i$ pixels. Next, we modify these probabilities with $q_i = \max q_{j \le i}$ to ensure the monotonicity. Since the pixel $i$ is not likely to be the supporting one if $q_i$ already approaches the maximum

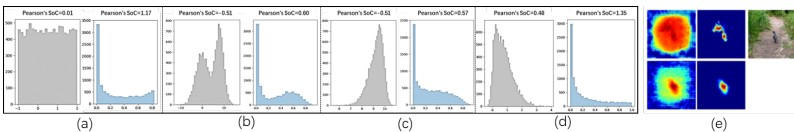

Figure 3: (a-d). The effect of CT on different distributions. (e) The effect of post hoc tuning on masks.

$q_{\max}=\max_i q_i$, we post hoc tune the relevance score as $r'_i=r_i\times v_i$, where $v_i=(q_{\max}-q_i)+\text{eps}$, and $r_i$ is original predicted score of the $i$-th pixel. An example is shown in Fig.3(e), where post hoc tuning is performed on the masks that obtained from BT (the top two) and CT (the bottom two).

Of note, benefiting from the enforced monotonicity, the ranking of relevance scores can be remained. Besides, we uniformly sample a limited number pixels for estimating the probabilities and infer the remaining with linear interpolation.

### 3.2.2 OPTIMIZATION WITH CLASSIFICATION LOSS

In this section, we introduce classification loss to optimize the above mask predictor. Denote $\ell_{\text{Cl}}$ as the classification loss used for training the black-box classifier $f$, the proposed predictor can be optimized without the non-trivial hyper-parameter tuning as

$$\operatorname{argmin}_{\mathcal{G}}\ell_{\text{Cl}}(f(\phi(\mathbf{I},\mathbf{M})),c), \text{ where } \mathbf{M} = \text{upsample}(\mathcal{C}(\mathbf{M}^{\text{g}})). \qquad (8)$$

Since the predictor is formulated to find the region that maximizes the target classification under the expected distribution, it can also better mimics the black-box classifier.

It is observed that when the cross entropy loss is used for training the original classifier $f$, Eq.8 is similar to removing all ad hoc constraints in Eq.2.

Thus, one may argue that the abandon of these constraints may remarkably scarify the quality of masks. For this, we provide an example of the ablation study by adding all constraints back and then display the masks from the trained predictors in Fig.2(b) under the following hyper-parameter settings: (1) carefully-tuned $\lambda$s, (2) $\lambda_{\text{av}}=0$, (3) $\lambda_{\text{av}}=\lambda_{\text{bg}}=0$, (4) $\lambda_{\text{av}}=\lambda_{\text{tv}}=0$, (5) $\lambda_{\text{av}}=\lambda_{\text{bg}}=\lambda_{\text{tv}}=0$. In general, we still obtain satisfying masks without any constraint. Detailed analysis is left to Sec. 4.4.

## 4 EXPERIMENTS

This section investigates the performance of the proposed method. We firstly evaluate the faithfulness and explainability with quantitative metrics. Then we perform the visualization of masks for further comparison. Finally, we introduce the case study to empirically demonstrate the effectiveness of two proposed innovations, respectively.

### 4.1 SETUP

In experiments, to demonstrate the broad applicability, we applied our method to CNNs with three types of structures. Besides, we compare Ours(BT) and Ours(CT) with nine recently proposed explanation methods. The experiments are performed on a real-world dataset, ImageNet. A brief introduction is given below, and details can be found in A.3.

**Three types of CNNs.** We use ResNet50 (He et al., 2016), VGG19 (Simonyan & Zisserman, 2014), and GoogleNet (Szegedy et al., 2015) as black-box classifiers to show the broad applicability. We load the parameters from traditional CNNs that were pre-trained using 1.2M ImageNet images.

**Nine compared methods.** The following methods are used for comparison: (1) Mask Generator (MGnet) (Dabkowski & Gal, 2017), (2) Meaningful Perturbation (MPert) (Fong & Vedaldi, 2017), (3) Grad CAM (GCAM) (Selvaraju et al., 2017), (4) Grad CAM++ (GCAM++) (Chattopadhay et al., 2018), (5) Feature Inversion (FInv) (Du et al., 2018b), (6) Extreme Perturbation (XPert) (Fong et al., 2019), (7) Vanilla Gradient (VGrad) (Simonyan et al., 2014), (8) Smoothness Gradient (SM-Grad) (Smilkov et al., 2017), (9) Integrated Gradient (ITGrad) (Sundararajan et al., 2017). In par-

ticular, all perturbation-based methods apply the same strategy to ours for adding noise. For the hyper-parameters in these methods, we follow the setting in their papers for a fair comparison.

**Implementation details.** We build the mask generator with three bottleneck blocks, which takes $7 \times 7$ feature maps as the low-level input, and predicts the mask at the $56 \times 56$ in size. We use a two-stage scheme to train the mask predictor. We first train the feature filter based on 250,000 images sampled from the training set, and then optimize other parts of the mask predictor with the batch size of 64. Of note, no ground truth is introduced and only the outputs of the classifier are utilized. We use Adam for 10 epochs with the initialized learning rate of $10^{-2}$. We apply step decay, and reduce the learning rate by half every three epochs. In addition, during training stage, 50% of cases the image $\mathbf{A}$ is the Gaussian blurred version of $\mathbf{I}$ with a variance of 10. The remainder of cases, $\mathbf{A}$ is set to a random colour image with the addition of a Gaussian noise.

**Quantitative metrics.** Evaluating the quality of masks based on a heuristical segmentation will reduce the fairness of comparisons. It is understandable that the segmentation will be significantly affected by thresholds (Fong & Vedaldi, 2017). To address this issue, we introduce two generalized metrics to evaluate the comprehensive quality in terms of both the faithfulness and the explainability.

*Faithfulness.* The explanation is expected to accurately replicate the model's behaviour. Here we rely on the smallest sufficient region of the image that alone allows a confident and consistent classification. Since it is subjective to decide how much confidence is preferred for a specific decision, we extend the evaluation into a general case by taking advantage of ranked pixels.

Specifically, we first estimate the class probability of a clean image as $q_{\mathrm{O}}$. Then we iteratively add $i$ ranked pixels with decreasing scores and measure the resulting probability $q_i$ of the target class, until the fully-perturbed images $q_{\mathrm{F}}$. Then, a preserving curve can be generated. The increased area under the curve with the intervals of $\Delta$ is used as a measure of faithfulness, calculated as

$$\mathcal{M}_{\mathcal{F}} = \sum_i \left( \frac{q_i}{q_{\mathrm{O}} - q_{\mathrm{F}}} \cdot \Delta \right) \times 100\%. \tag{9}$$

An extra discussion between $\mathcal{M}_{\mathcal{F}}$ and perturbation-based methods are provided in A.4.

*Explainability.* For quantitative evaluation of explainability, we generally use the extra information such as bounding boxes. For example, in the pointing game, the maximum point is first extracted from each generated mask. Then, according to whether the maximum point falls in one of the ground truth bounding boxes or not, a hit $N_{\mathrm{Hits}}$ or a miss $N_{\mathrm{Misses}}$ is counted. The localization accuracy of the pointing game for each object category is defined as:

$$\mathcal{M}_{\mathcal{PG}} = \frac{N_{\mathrm{Hits}}}{N_{\mathrm{Hits}} + N_{\mathrm{Misses}}} \times 100\%. \tag{10}$$

This process is repeated for all categories and the results are averaged as the final accuracy. However, $\mathcal{M}_{\mathcal{PG}}$ only unitizes one pixel for evaluation, leading to a large bias. To address this issue, weakly-supervised object localization considers a boarder pixels by calculating the intersection over union between binary masks and images. Nevertheless, it still faces the issue of the choice of thresholds.

Thus, we introduce a new metric by regarding relevance scores as the results of retrieval tasks. Specifically, we denote the precision as the fraction of the pixels retrieved within bounding boxes. Recall denotes the fraction of the within-bounding-box pixels that are successfully retrieved. By computing a precision $P_i$ and recall $R_i$ at each position in the ranked scores of pixels, we can evaluate the explainability of scores by the area under the precision-recall curve:

$$\mathcal{M}_{\mathcal{I}} = \sum_i P_i \cdot (R_i - R_{i-1}) \times 100\%. \tag{11}$$

## 4.2 Quantitative Evaluation

**On the faithfulness with $\mathcal{M}_{\mathcal{F}}$.** To evaluate the faithfulness of the proposed predictor, we calculate the mean metric $\mathcal{M}_{\mathcal{F}}$ based on 10,000 sampled images from the validation set. We set $\Delta = \frac{1}{32}$ as the interval. Besides, we simply add a smoothed mask over the original one with a tiny weight, introducing the ranking for pixels with zero scores. We perturb images with their Gaussian blurred version based on their masks and estimate the average faithfulness. In order to avoid outliers, we introduce min-max normalization on $\mathcal{M}_{\mathcal{F}}$s over all methods for each image. Of note, although post hoc tuning remains the ranking of pixels, it is not performed for avoiding the ambiguity.

| | $\mathcal{M}_{\mathcal{F}}$ | | | | $\mathcal{M}_{\mathcal{I}}$ | | | | $\mathcal{M}_{\mathcal{PG}}$ | | | |
|---|---|---|---|---|---|---|---|---|---|---|---|---|
| | ResNet | VGG | Google | MEAN | ResNet | VGG | Google | MEAN | ResNet | VGG | Google | MEAN |
| Ours(BT) | 77.32 | 80.10 | 75.68 | 77.70 | 83.40 | 82.69 | 81.46 | 82.52 | 87.92 | 79.37 | 88.63 | 85.31 |
| Ours(CT) | 81.42 | 79.49 | 73.23 | **78.05** | 83.02 | 83.91 | 81.73 | 82.89 | 85.28 | 87.45 | 88.97 | 87.23 |
| MGnet | 70.68 | 68.52 | 56.54 | 65.25 | 82.62 | 83.53 | 81.87 | 82.67 | 89.13 | 89.37 | 89.40 | **89.30** |
| MPert | 47.52 | 49.04 | 41.56 | 46.04 | 76.82 | 74.80 | 72.26 | 74.63 | 82.39 | 80.34 | 78.55 | 80.43 |
| FInv | 71.34 | 65.91 | 63.35 | 66.87 | 75.20 | 72.63 | 75.35 | 74.39 | 81.95 | 80.47 | 79.64 | 80.69 |
| XPert | 65.13 | 64.06 | 48.91 | 59.37 | 78.73 | 80.92 | 71.26 | 76.97 | 77.72 | 79.38 | 77.31 | 78.14 |
| GCAM | 75.43 | 67.08 | 67.66 | 70.06 | 79.28 | 74.93 | 82.15 | 78.79 | 78.33 | 84.67 | 88.94 | 83.98 |
| GCAM++ | 74.18 | 73.63 | 64.69 | 70.83 | 82.86 | 84.46 | 83.20 | **83.51** | 84.11 | 90.30 | 89.23 | 87.88 |
| VGrad | 13.88 | 15.83 | 23.12 | 17.61 | 66.23 | 70.90 | 66.78 | 67.97 | 77.71 | 88.08 | 85.64 | 83.81 |
| SMGrad | 38.34 | 38.93 | 42.16 | 39.81 | 73.17 | 74.03 | 70.31 | 72.50 | 88.61 | 88.67 | 89.35 | 88.88 |
| ITGrad | 31.47 | 50.80 | 52.02 | 44.76 | 66.92 | 66.74 | 63.25 | 65.64 | 83.47 | 89.39 | 87.32 | 86.73 |

Note: For simplicity, ResNet, VGG and Google stand for ResNet50, VGG19 and GoogleNet, respectively.

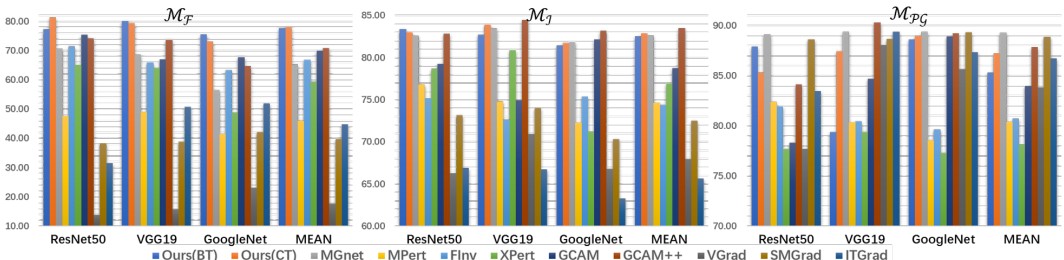

Figure 4: The quantitative results of eleven methods on three CNNs. (left) Performance on $\mathcal{M}_{\mathcal{F}}$, (middle) Performance on $\mathcal{M}_{\mathcal{I}}$, (right) Performance on $\mathcal{M}_{\mathcal{PG}}$. Detailed values are listed in the above table.

The results of average $\mathcal{M}_{\mathcal{F}}$ are displayed in Fig.4(left). Based on the results, the following observations can be obtained. Firstly, VGrad, SMGrad, and ITGrad are generally worse than others with a large gap. It is understandable that, these three methods only search sensitive pixels with gradients, the pixels with high scores will discretely appear in each image. As a result, these methods become harder to gather the sufficient supporting information in a local receptive field and reach a high class probability. Secondly, GCAM obtains comparable performance to GCAM++. Besides, FInv that regularizes masks by utilizing intermediate layers outperforms MPert. It suggests that, without using the lowest resolution feature maps, an effective regularization is of importance for individually optimizing the mask of each instance. Thirdly, our methods outperform most compared methods, and obtain much higher performance than MGnet. Consider that the three methods apply the same architecture of mask generators. Thus, the results demonstrate that guiding the relevance scores towards right-skewed distributions enjoys a better ranking of supporting features.

**On the explainability with $\mathcal{M}_{\mathcal{I}}$.** To reveal the explainability, we first introduce $\mathcal{M}_{\mathcal{I}}$ to evaluate the performance of object location. For this, we resize and crop bounding boxes to the size of $224 \times 224$, leading to the same size of test images. The experiments are performed on 10,000 validation images with bounding box annotations.

The results of average $\mathcal{M}_{\mathcal{I}}$ are displayed in Fig.4(middle). From these results, we have the following observations. Firstly, GCAM++ consistently obtained much better performance than GCAM. It is understandable that the former is able to detect multiple objects in the image and assign them high relevance scores. Secondly, by replacing all constraints with a simple distribution controller, Ours(CT) outperforms MGnet with a small gap, and both of them enjoy better performance than most of other methods. The reason is that, benefiting from the training with large-scale images, their predicted high relevance scores focus on objects more robustly. Thirdly, the last three gradient-based methods perform worse than others. The possible reason is that, gradients are insensitive to the smooth supporting regions, which makes these regions ignored and reduces the performances.

**On the explainability with $\mathcal{M}_{\mathcal{PG}}$.** In this experiment, we show that our methods still bring comparable performances in pointing games. The average results of all compared methods on pointing games are shown in Fig.4(right). From the results of this figure, the following observations can be obtained. Firstly, SMGrad and ITGrad now obtain comparable or even better performance than the first eight methods and outperform VGrad. The main reason is that, this metric only considers the localization precision of the top pixel rather than a group of pixels. By removing noise in gradients to some degree, these two methods are able to locate the most important pixel. Secondly, although our methods remove all constraints, e.g, $\lambda_{\mathrm{bg}}$=0, and $\lambda_{\mathrm{tv}}$=0, they still obtain the comparable accuracy

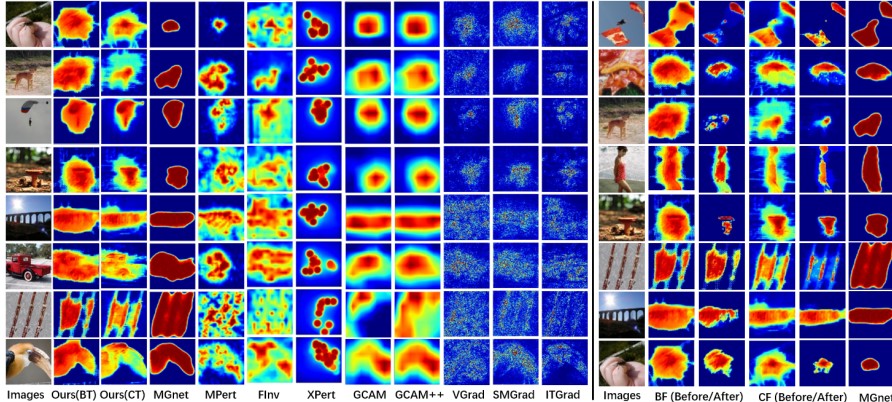

Figure 5: (left). Masks of different methods with ResNet50. (right). The effect of post hoc tuning.

of locating the most important pixel. One possible reason is that, by optimizing the predictor with training data, a large number of noisy features can be removed, leading to the robust estimation of scores. Thirdly, MGnet enjoys the best performance. It means that, by enforcing the background towards a low probability, the high scores on the background are generally prevented.

### 4.3 EXPLAINABILITY WITH VISUALIZATION

**Comparison to baseline methods.** We qualitatively compare the saliency maps produced using our methods with those produced by other baselines. The red and blue denotes the high and low score, respectively. In this experiments, no post hoc tuning is used for the proposed methods. We randomly sample images from ImageNet and show their masks obtained with ResNet50 in Fig.5(left). More examples from other CNNs are displayed in Figs.8-12 in A.5.

From these results, we have the following observations. Firstly, the gradient-based methods generally results in more high scores outside the objects. Secondly, although the high-level feature maps are used to build saliency maps, GCAM and GCAM++ may still miss the supporting pixels of objects, such as the last second row in Fig.5(left). Thirdly, MGnet obtains high scores on the supporting features. However, it only assigns them similar relevance scores. Finally, compared with other methods, with an easy setting of hyper-parameters, Ours(CT) not only predicts higher scores on these pixels than others but also builds discriminative scores over these pixels. It shows the effectiveness of guiding scores under right-skewed distributions.

**The effect of post hoc tuning.** In this experiment, we aim to investigate the effectiveness of post hoc tuning to improve the distribution of relevance scores. Fig.5(right) shows same examples of our masks with ResNet50 before and after the tuning operation. The masks of MGnet are also displayed for comparison. More examples of this task can be found in Fig.13 in A.5.

From the figure, it is observed that compared with MGnet, our predicted masks sacrifice the difference of scores between the supporting features and others to some degree. However, owing to the better ranking of scores over pixels (which is empirically demonstrated by the metric $\mathcal{M}_\mathcal{F}$), we can easily utilize the post hoc tuning to remove the noise and highlight the supporting pixels.

### 4.4 PERFORMANCE OF THE INNOVATIONS IN OUR PREDICTOR

**On distribution controller: the effect of distributions.** To show that the distributions of output masks are benefited from the proposed controller, we study the masks from the predictors optimized under varying preset distributions. Specifically, we change the hyper-parameters of controllers with $\{(\eta,h)\}=\{(0.5,2.5),(1.5,2.5),(1.5,1.5),(2.5,2.5),(1.5,0.5)\}$. Then we train mask predictors with these controllers. The resulted masks with the corresponding preset distributions are shown in Fig.6(left).

As we observed, although it is hard to expect that all masks obtain scores with the same distribution to the expected one, the controllers enforce them towards these distributions. Besides, we perform the transformer on the obtained masks of MGnet in a post hoc manner, which fails to build discrim-

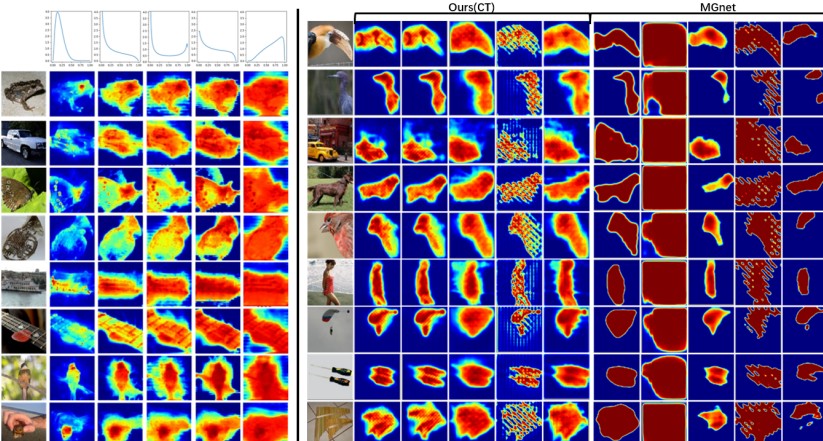

Figure 6: (left) The effect of preset distributions (1st row) on the predicted masks (2nd-8th rows). (right) The ablation study on the regularizations of different methods.

inative scores on supporting features. The reason is understandable that the scores are already be stacked at the values of 0 and 1.

**On optimization with classification loss: the effect of regularization.** To evaluate the impact of our simplified objective function, we introduce an ablation study to analyze how the regularization terms affect our mask predictor. We add all ad hoc constraints back to the formulation and train the mask predictor using the following hyper-parameter settings: (1) carefully-tuned $\lambda$s, (2) $\lambda_{av}$=0, (3) $\lambda_{av}$=$\lambda_{bg}$=0, (4) $\lambda_{av}$=$\lambda_{tv}$=0, (5) $\lambda_{av}$=$\lambda_{bg}$=$\lambda_{tv}$=0. For comparison, an ablation study is also perform on MGnet with the setting of (1) carefully-tuned $\lambda$s, (2) $\lambda_{av}$=0, (3) $\lambda_{bg}$=0, (4) $\lambda_{tv}$=0, (5) $\lambda_{bg}$=$\lambda_{tv}$=0. The masks from the trained models with varying settings are displayed in Fig.6(right).

By adding all ad hoc constraints back, the masks of Ours(CT) can be improved to some degree. However, we observed that $\lambda_{av}$=0 has few effect on our predictor. In addition, $\lambda_{bg}$=0 will increase the noise around the boundary owing to the smoothness constraint, and $\lambda_{tv}$=0 causes the holes in the masks. Nevertheless, by further removing these two terms, these shortcomings can be alleviated and the final masks of Ours(CT) still remain satisfying quality. On the contrary, besides $\lambda_{tv}$=0, all terms result in remarkable impacts on the masks of MGnet. Of note, although $\lambda_{bg}$=0 can release the mask towards discriminative scores, it would reduce the discrimination on the majority of supporting features, such as the second, fourth and fifth rows. Besides, it increases the sensitiveness of $\lambda_{av}$ on masks. For example, while $\lambda_{bg}$ is carefully set, $\lambda_{av}$ varying within (2,12) consistently leads to acceptable masks for ResNet50. Once $\lambda_{av}$=0, the number of high scores is totally controlled by $\lambda_{av}$ and the quality of masks is only acceptable with $\lambda_{av} \in$(2,4). In short, benefiting from the distribution controller, Ours(CT) is insensitive to the regularization terms and can predict satisfying masks without ad hoc constraints.

## 5 CONCLUSION AND FURTHER WORK

This paper presents a simple but effective mask predictor to provide local explanations for black-box classifiers. Specifically, we introduce the conception of distribution controller on relevance scores, and integrate it with a trainable mask generator to directly guide the relevance scores. In addition, the proposed predictor is optimized under the classification loss, aiming to better mimic the classifiers. Based on the proposed innovations, our method enables discriminative scores over supporting features and also facilitates the setting of involved hyper-parameters. The experimental results demonstrate that our method outperforms others in terms of faithfulness and explainability.

There are some aspects needing further investigations. Firstly, since the objective functions are formulated the same to those of classifiers, it is natural to train the classifiers and mask predictors at the same time. As a result, we can provide users a pair of classifier and explainer. Secondly, although this paper provides an intuitive comparison of the transformed distributions for setting hyper-parameters, a quantitative analysis on the proportion of features at the tail could be studied.

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

# A    APPENDIX

## A.1    THE FEATURE FILTER IN THE MASK GENERATOR

The purpose of the feature filter is to attenuate spatial locations whose contents do not correspond to the selected class. Denote $\mathbf{X} \in \mathbb{R}^{H \times W \times C}$ as the output of the last convolution layer of the classifier, where $H, W$ $C$ indicate the height, the width, and the number of channels of its feature maps, respectively. The output of filter $\mathbf{Y} \in \mathbb{R}^{H \times W \times C}$ at the spatial location $i, j$ is formulated as

$$\mathbf{Y}_{ij} = \mathbf{X}_{ij} \text{sigmoid}(\mathbf{X}_{ij}^{\mathrm{T}} \mathbf{C}_s), \tag{12}$$

where $\mathbf{C}_s \in \mathbb{R}^{1 \times C}$ is the embedding of the class $c$. For efficient optimization, (Dabkowski & Gal, 2017) introduces noise on real labels. Specifically, gradient-based optimizers are employed to maximize $\text{mean}(\text{sigmoid}(X_{ij}^T C_k))$, $k=s$ and minimize $\text{mean}(\text{sigmoid}(\mathbf{X}_{ij}^{\mathrm{T}} \mathbf{C}_s))$, $k \neq s$, iteratively.

## A.2    THE SETTING OF THE HYPER-PARAMETERS OF CT

Now we show the way of easily setting the hyper-parameters of CT based on the intuitive geometric properties of probability density functions.

To do this, we estimate the probability density function of $m^{\mathrm{d}}$. With the probability density transformation Forbes et al. (2011), the transformed probability density function can be obtained as

$$p(m^d) = \frac{1}{\sqrt{2\pi}h\eta} \cdot \frac{1}{m(1 - m^{1/h})} \cdot \exp\left(-\frac{(\ln(m^{(-1/h)} - 1))^2}{2\eta^2}\right), \tag{13}$$

where the superscript $d$ is removed for clarity. The detailed proof is provided at the end of A.2. Now we set the hyper-parameters based on their effects on the geometry of probability density functions.

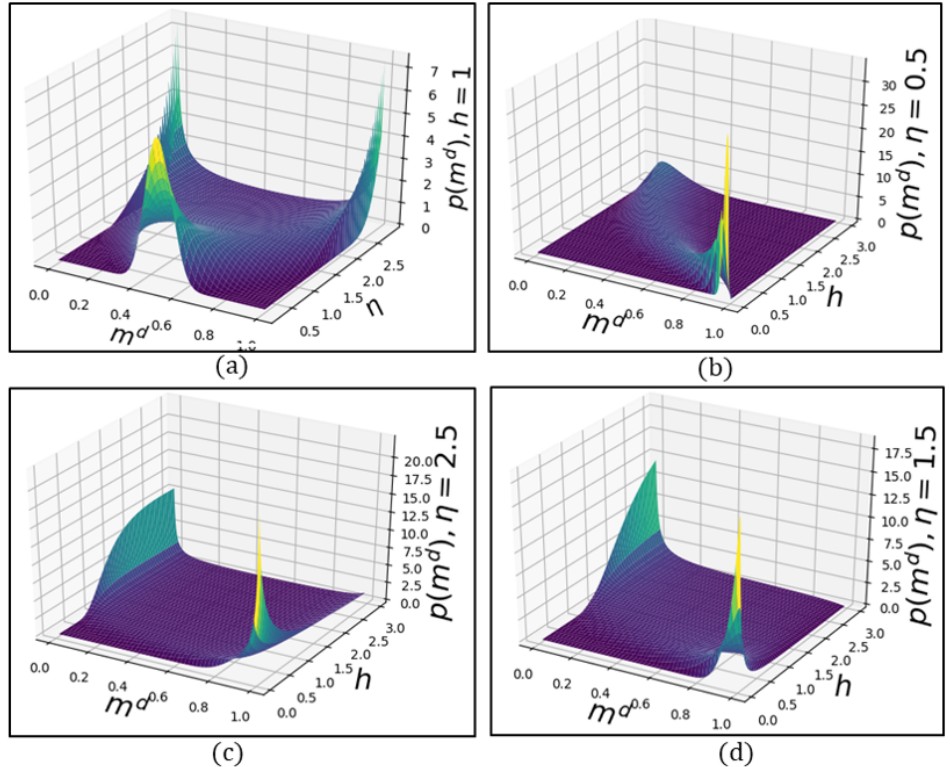

Figure 7: The probability density functions with the different settings of hyper-parameters.

We fix $h$=1 and observe the effect of $\eta$. The corresponding probability density function is displayed in Fig.7(a). By changing $\eta$ within (0.5,2.5), the density functions of $p(m^d)$ approximately change from the concave to the convex for $m^d \in (0,1)$. In particular, $\eta$=1.5 approximately leads $p(m^d)$ to an uniform distribution. Considering $1.5 \sim 0.9\sqrt{\pi}$, it is also consistent to the sigmoid approximation of the cumulative probabilities of the standard normal distribution (Waissi & Rossin, 1996).

Then we set $\eta$={0.5,1.5,2.5} and observe the effect of $h$. Three probability density functions are displayed in Figs.7(b-d). As we can see, $p(m^d)$ with $\eta$=0.5 leads to extremely low probabilities for $m^d$>0.5, which will potentially waste this range of scores for highlighting supporting features. Besides, owing to the strong convexity in the first step of the transformer, $p(m^d)$ with $\eta$=2.5 continues this undesired property. On the contrary, benefiting from an approximated uniform distribution with $\eta$=1.5, the distribution of $p(m^d)$ in Fig.7(d) is modified from the trend of the left-skewed to the right-skewed with a light tail over $(0,1)$. Since a proportion of features corresponding to the area at the tail will be highlighted with higher scores, we fix $\eta$=1.5 and $h$=2.5 to obtain discriminative scores on supporting features.

**Proof.** According to the probability density transformation Forbes et al. (2011), the transformed probability density function can be obtained based on that of the original variable:

$$p(m) = p_z(g_z^{-1}(m)) \cdot |\frac{\partial g_z^{-1}(m)}{\partial m}|, \tag{14}$$

where $p_z(\cdot)$ is the probability density function of the original variable $z$, and $g_z^{-1}(m)$ is the inverse function of $m$ on $z$. Since $m = g_z(z) = (\frac{1}{1+\exp(-\eta z)})^h \in (0,1)$ in Eq.7, we obtain

$$z = g_z^{-1}(m) = -\frac{1}{\eta}\ln(m^{-1/h} - 1), \tag{15}$$

where $(m^{-1/h}\text{-}1)>0$. Besides, we obtain:

$$
\begin{aligned}
|\frac{\partial g_z^{-1}(m)}{\partial m}| &= \frac{1}{\eta} \cdot \frac{1}{m^{(-1/h)} - 1} \cdot (\frac{1}{h} m^{(-1/h)-1}) \\
&= \frac{1}{\eta h} \cdot \frac{1}{m} \cdot \frac{m^{(-1/h)}}{m^{(-1/h)} - 1} \\
&= \frac{1}{\eta h} \cdot \frac{1}{m(1 - m^{(1/h))}}
\end{aligned}
\tag{16}
$$

Recall the probability density function of a normal distribution is

$$
p_z(z) = \frac{1}{\sqrt{2\pi}\sigma} \exp(-\frac{z^2}{2\sigma^2}).
\tag{17}
$$

where $\sigma=1$ in standard normal distributions. By substituting Eqs.15-17 into Eq.14, we obtain

$$
p(m) = \frac{1}{\sqrt{2\pi}h\eta} \cdot \frac{1}{m(1 - m^{(1/h)})} \exp(-\frac{(\ln(m^{(-1/h)} - 1))^2}{2\eta^2}),
\tag{18}
$$

which completes the proof.

### A.3 A DETAILED INTRODUCTION OF THE EXPERIMENTAL SETTING

Firstly, we present a brief description on the experimental settings of compared methods.

- Mask Generator (Dabkowski & Gal, 2017), which introduces the multiple constraints into the objective functions for training the predictor. For this method, we use the same setting in our method and using its default hyper-parameters in its publicly available codes.
- Meaningful Perturbation (Fong & Vedaldi, 2017), which performs meaningful image perturbations and directly optimizes masks with designed constraints. For comparison, we use the same kind of perturbations in our method, and apply an Adam optimizer with the learning rate of 0.1 for optimization. The iterations is set to 300.
- Grad CAM (Selvaraju et al., 2017), which upsamples the saliency maps based on the gradient-weighted high-level feature maps. We use the last convolutional layers to build its coarse saliency map and upsample it to the image scale as the final saliency map.
- Grad CAM++ (Chattopadhay et al., 2018), which conducts a weighted combination of the positive partial derivatives to generate a visual explanation. We apply Grad CAM ++ with exponential functions for efficiency.
- Feature Inversion (Du et al., 2018b), which attempts to perform regularization on masks with guided feature inversion. Following their setting of hyper-parameters, we first optimize the masks to reconstruct the lowest resolution features based on the second lowest resolution for 10 iterations, and then optimize the masks under the class-discriminative interpretation for 70 iterations.
- Extreme Perturbation (Fong et al., 2019), which introduces extreme perturbation with an area constraint. We directly use the authors' pubic available codes for our experiments. For efficiency, the number of iteration is set to 300.
- Vanilla Gradient (Simonyan et al., 2014), the basic method that uses the gradients of the raw images as saliency maps. To improve the visualization, we crop outliers and normalize the scores to $[0, 1]$.
- Smoothness-Gradient (Smilkov et al., 2017), which removes noise by adding noise to images. We apply 20% noise as suggested and set the sample size to 50. Similarly, we crop outliers and normalize the scores.
- Integrated Gradient (Sundararajan et al., 2017), which combines the implementation invariance of gradients along with the sensitivity. We introduce black images as the baseline and set 200 as the number of steps in its Riemman approximation of the integral. We use the same way of improving visualization for saliency maps.

Then, we introduce the structures of mask generators for different CNNs. Specifically, for ResNet50 which consists of intermediate layers with $\{64,256,512,1024,2048\}$ channels at different scales,

we set three low-to-high-resolutions blocks in mask generators with the channels of $\{256,128,64\}$, namely, one quarter channels of the classifier; for VGG19 and GoogleNet that contain the layers with $\{64,128,256,512,512\}$ and $\{64,192,480,832,1024\}$ channels, we half the channels in mask generators, namely, $\{256,128,64\}$ and $\{416,240,96\}$ for these blocks, respectively.

## A.4 THE ANALYSIS ON THE RELATIONSHIP BETWEEN THE PROPOSED METRIC AND THE PERTURBATION-BASED METHODS

We provide a new view to discuss the relationship between the proposed metric and wildly used perturbation-based methods with the mask involved. Recall that in most perturbation-based methods, we aim to maximize the probability of the target class of the perturbed image:

$$\operatorname{argmin} f_c(\phi(\mathbf{I}, \mathbf{M})), \tag{19}$$

where the pixels with higher scores in masks are supposed to be more important. By decomposing the above mask into multiple ones with one-hot coded mask $\mathbf{M}_i$, Eq.19 is equal to

$$f_c(\phi(\mathbf{I}, \sum_{i=0}^{N} \alpha_i \mathbf{M}_i)) \tag{20}$$

where $\alpha_i$ denotes the relevance score with $\alpha_i > \alpha_{i+1}$. Suppose $\beta_i$ indicate some positive weights and denote $\alpha_i = \sum_{j=i}^{N} \beta_j$, the above equation can be further transformed into

$$f_c(\phi(\mathbf{I}, \sum_{i=0}^{N}(\sum_{j=i}^{N} \beta_j)\mathbf{M}_i)) = f_c(\phi(\mathbf{I}, \sum_{i=0}^{N}(\beta_i + ... + \beta_N)\mathbf{M}_i)). \tag{21}$$

Recall the area under the probability vs. the number of pixels, which can be formulated as

$$\sum_{j=0}^{N}(\beta_j f_c(\phi(\mathbf{I}, \sum_{i=0}^{j} \mathbf{M}_j))) = \beta_0 f_c(\phi(\mathbf{I}, \mathbf{M}_0))) + \beta_1 f_c(\phi(\mathbf{I}, \mathbf{M}_0 + \mathbf{M}_1)) + .... \tag{22}$$

We observe that Eq.21 tends to be a linear approximation of Eq.22. Specifically, two of them become equal when $f_c(\cdot)$ and $\phi(\cdot)$ are linear. According to Eq.22, the larger values on the former $\beta_i$ means the users pay more attention to the top features. According to Eq.21, it also implies the high scores should be much larger than others and highlight supporting features. Thus, to evaluate the effectiveness of perturbation-based methods, the proposed metric tends to be more convincing.

## A.5 SUPPLEMENTARY RESULTS ON VISUALIZATION EVALUATION

In this section, we provide the supplementary results on visualization evaluation.

Specifically, in Figs.8-12, we display the predicted masks obtained from different methods with ResNet50, VGG19, and GoogleNet. By compared the proposed methods with others in these figures, we observe that our methods are generally able to distinguish the objects from the backgrounds, and also obtain discriminative scores at different parts of objects. It thus provides better explanations for the behaviours of black-box classifiers.

Besides, to further demonstrate the effect of post hoc tuning, we show more masks before and after the tuning operation in Fig.13. As we can see, benefiting from the effective ranking of supporting pixels, we can avoid the noise outside of the objects and further improve the masks.

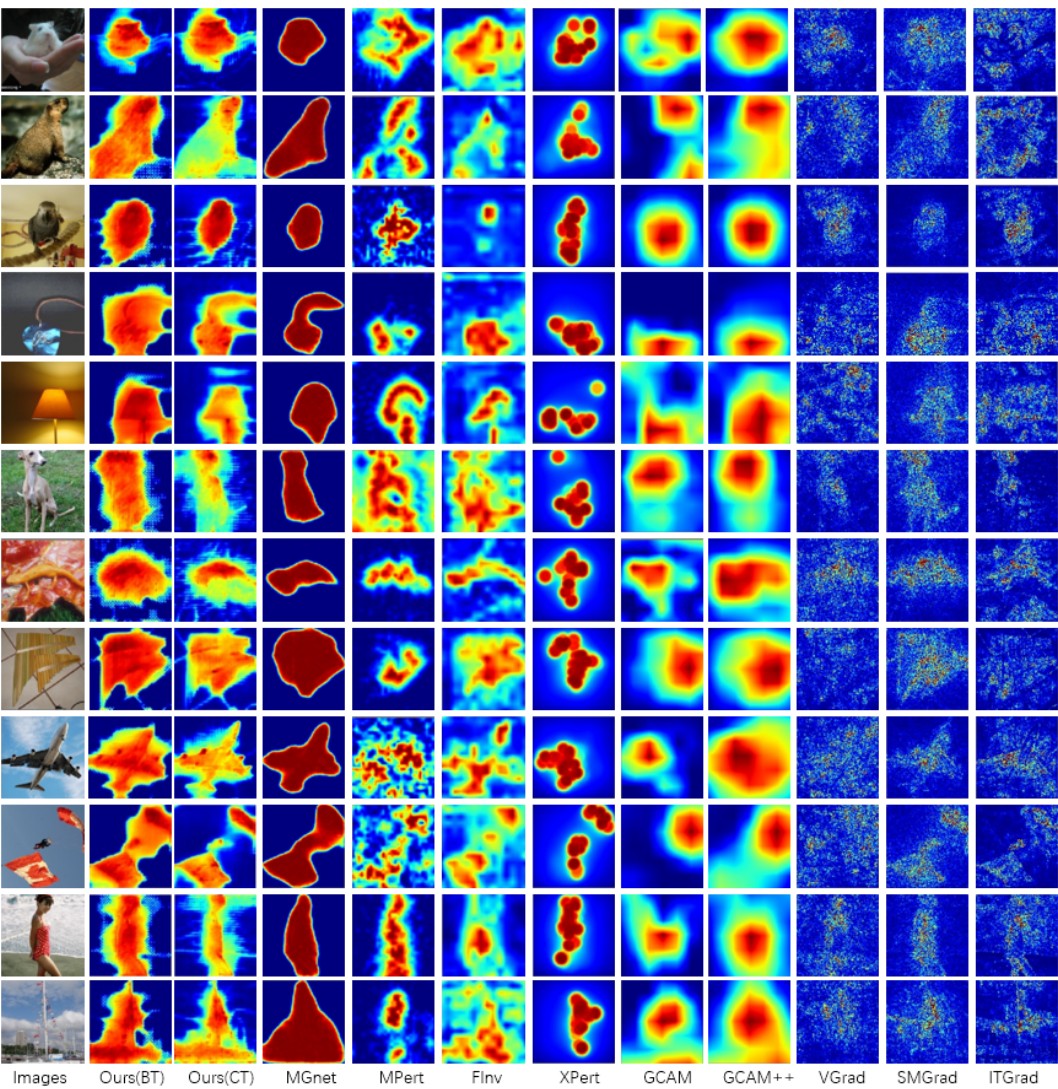

Figure 8: Examples of the saliency maps of ResNet50.

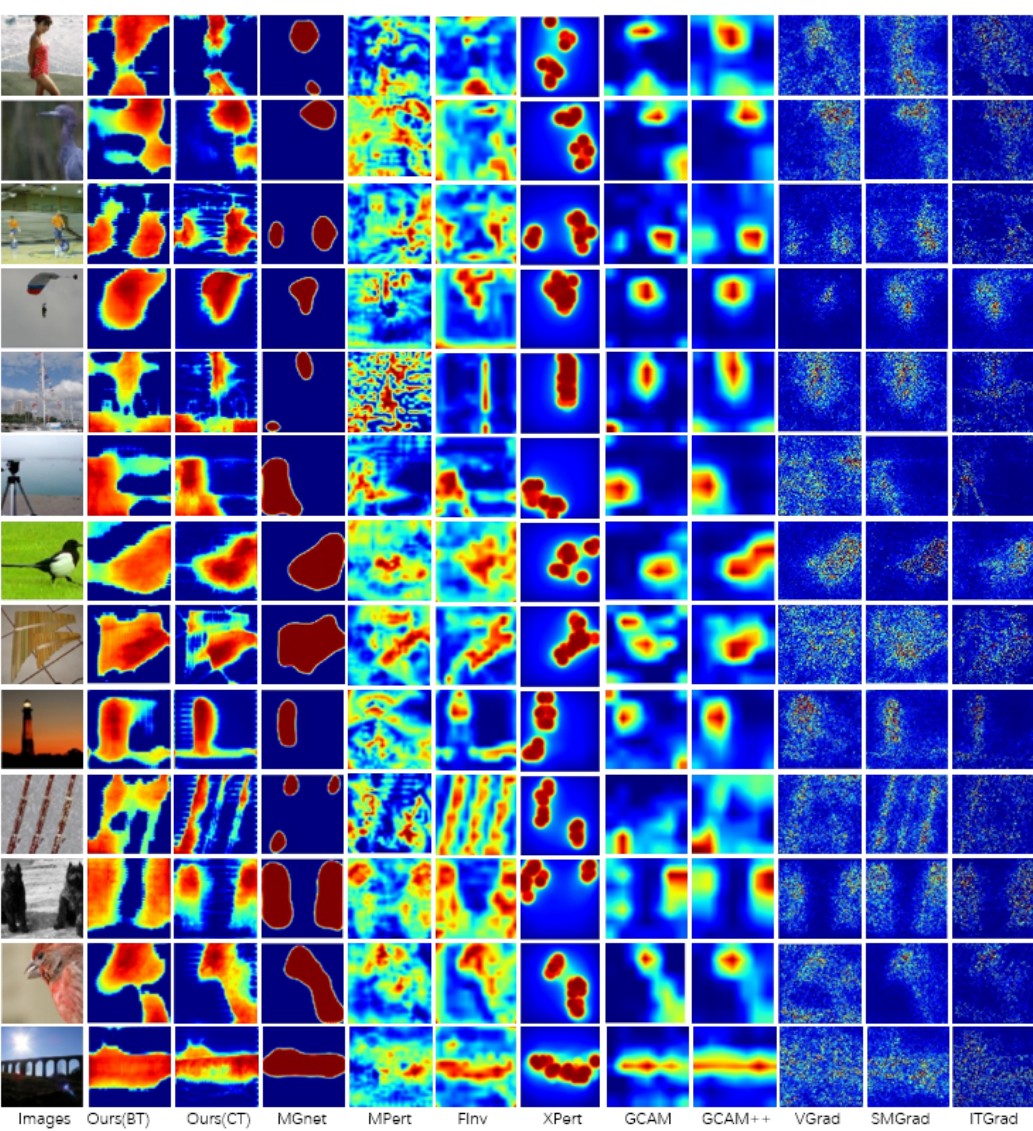

Figure 9: Examples of the saliency maps of VGG19.

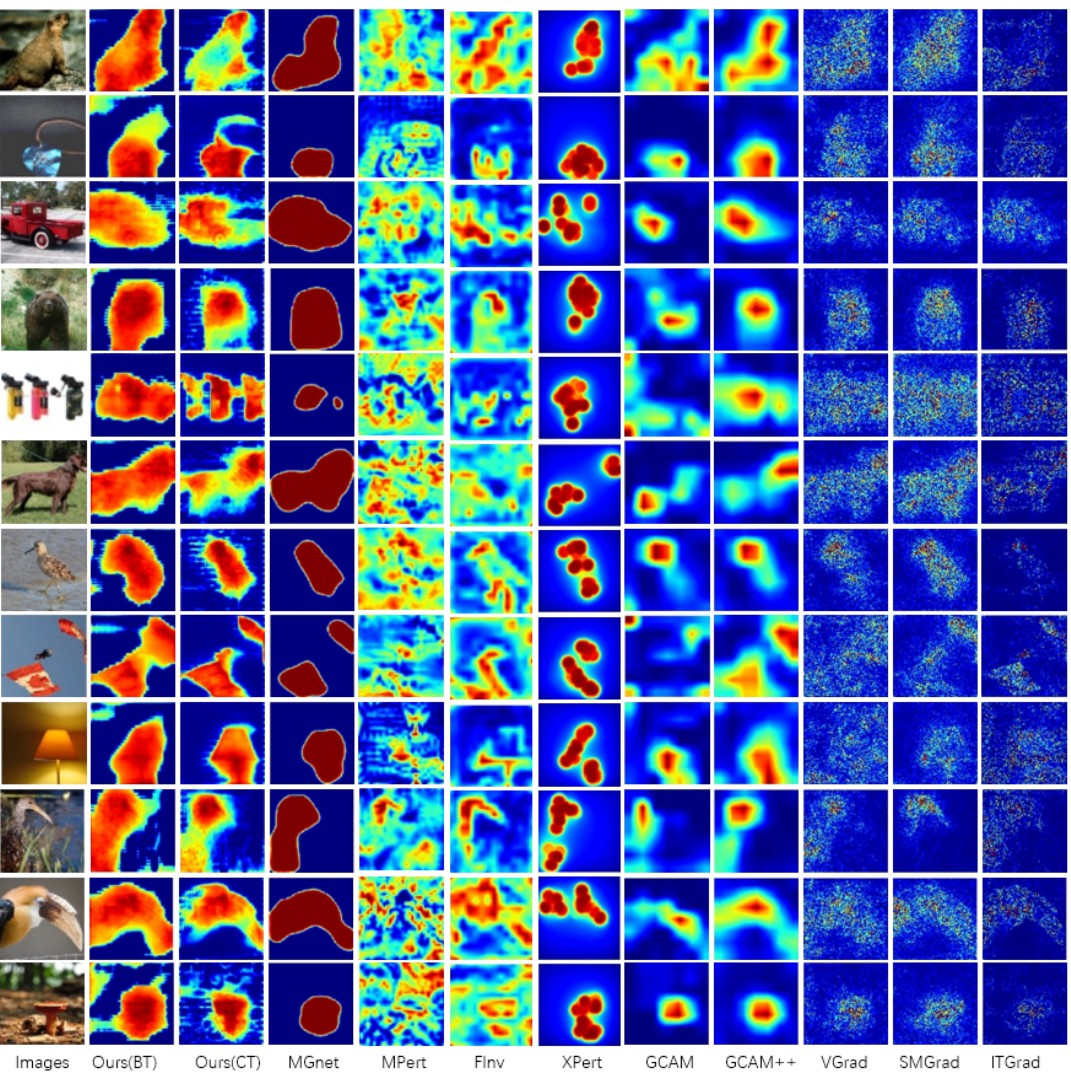

Figure 10: Examples of the saliency maps of VGG19.

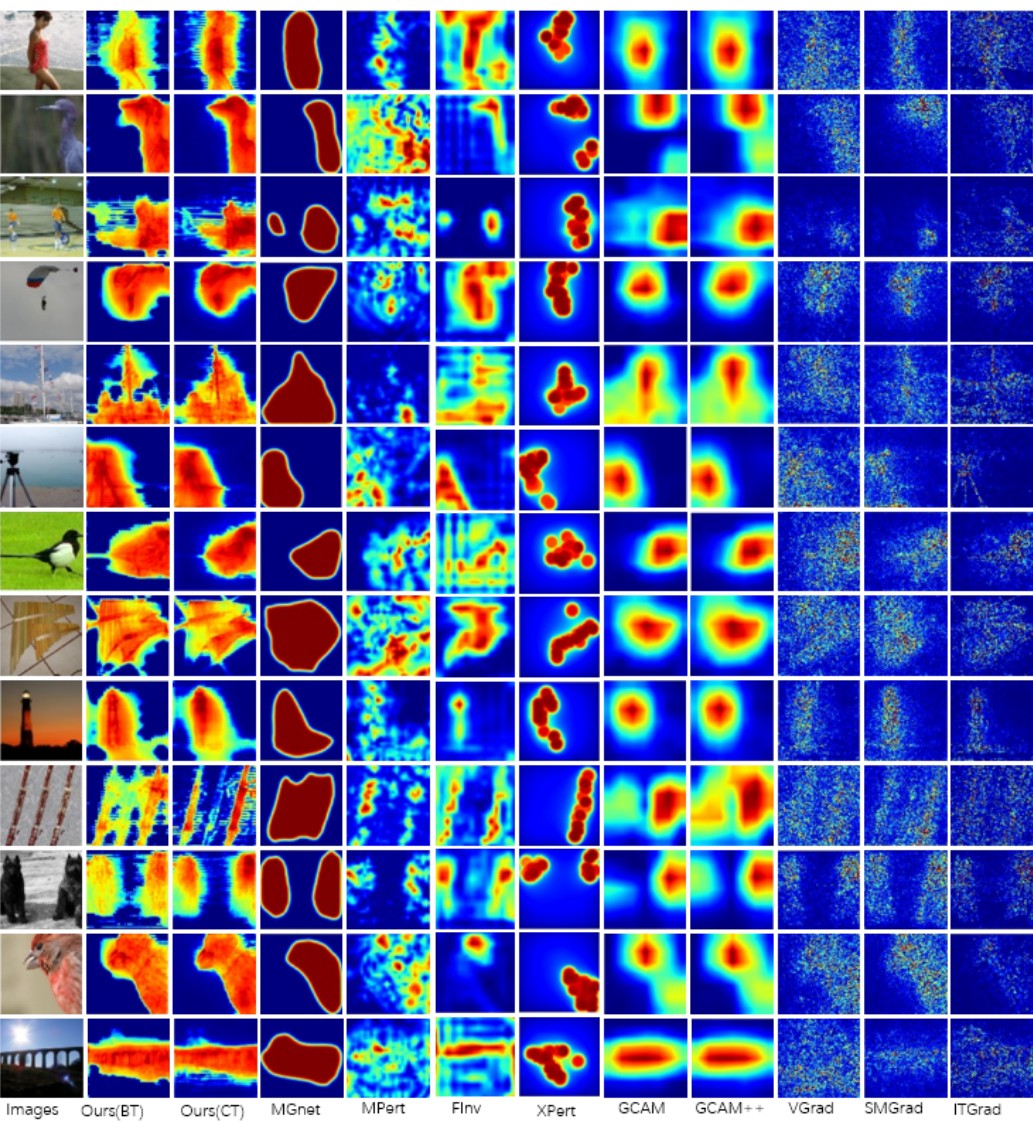

Figure 11: Examples of the saliency maps of GoogleNet.

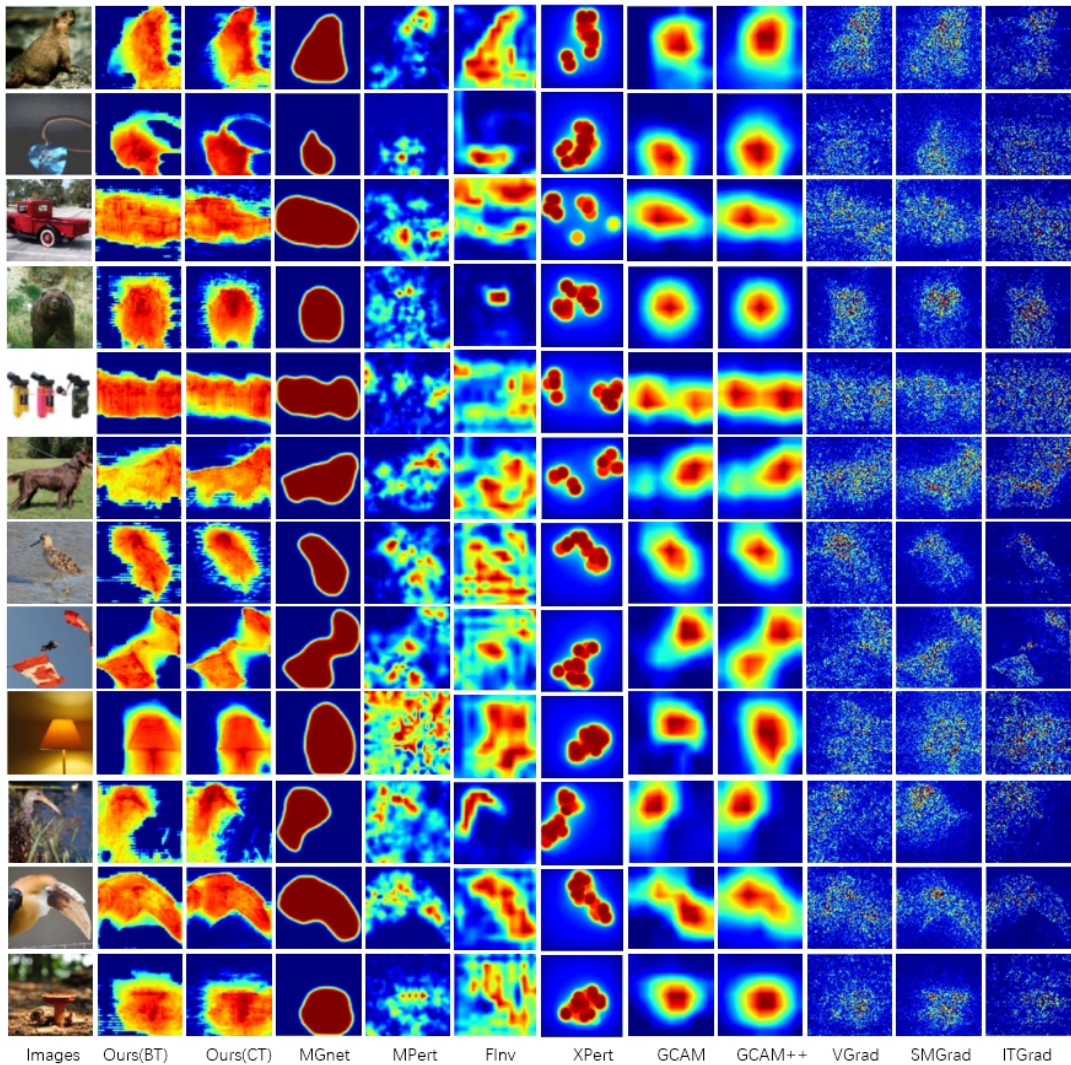

Figure 12: Examples of the saliency maps of GoogleNet.

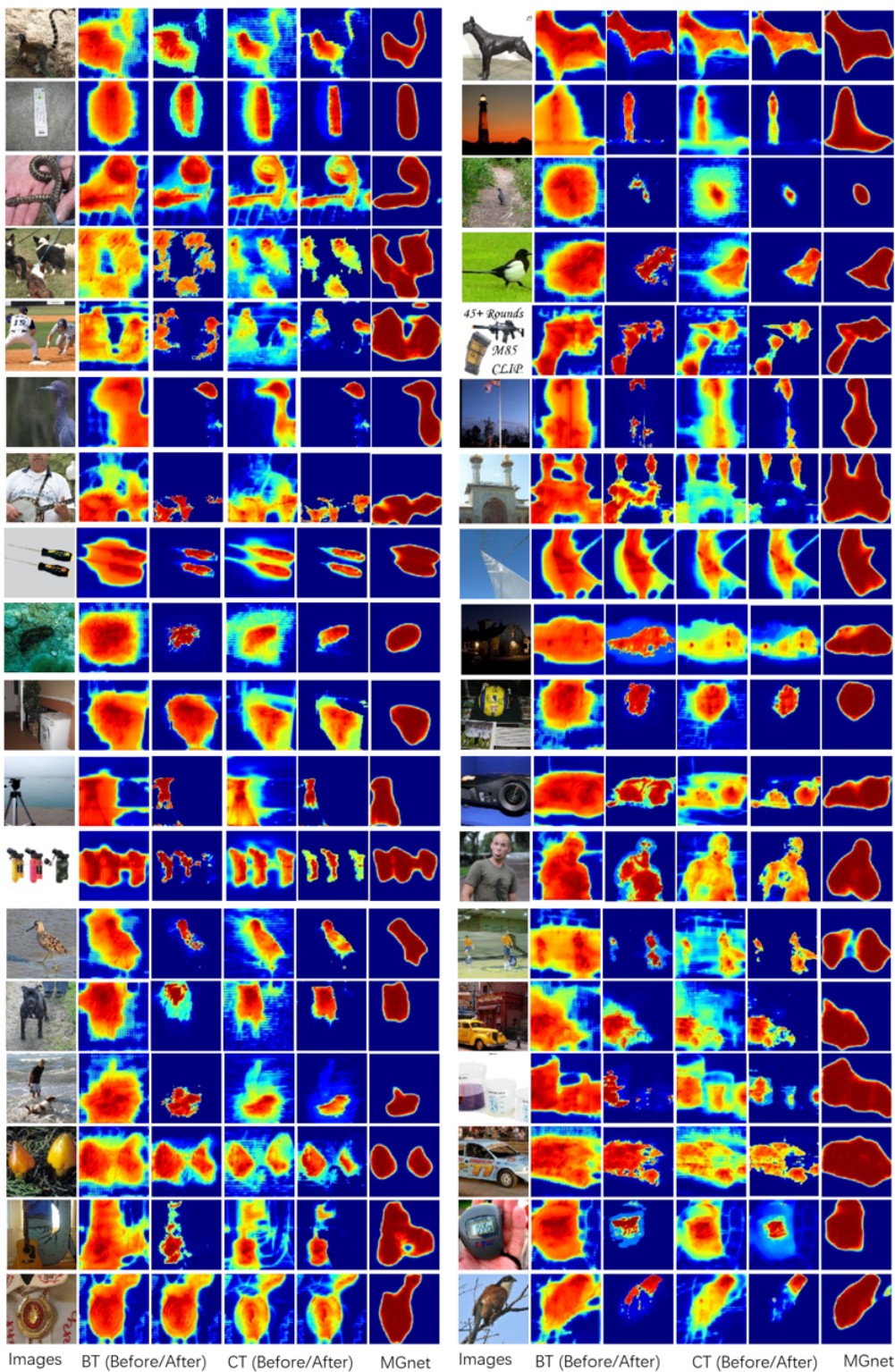

Figure 13: Saliency maps of the proposed methods before and after post hoc tuning.

