# OpenReview forum: "Distribution-Guided Local Explanation for Black-Box Classifiers"
_ICLR.cc/2020/Conference — Reject_

### Official Review · AnonReviewer1 · 2019-10-16
**Official Blind Review #1**

**Rating:** 6

**Review:**

This paper proposes a mask predictor to bridge the information gap between the classification loss and the constraints introduced to obtain satisfying explainability. The predictor consists of a distribution controller and a mask generator to refine the mask towards the desired score distribution. The predictor is optimized solely under the classification loss without additional constraints, which therefore improves the faithfulness of mimicking target black-box models. This paper introduces two metrics to evaluate the proposed mask predictor and the experiments demonstrate the proposed method outperforms others in terms of faithfulness and explainability.
Overall, this paper has some advantages:
(1) the method could be a significant contribution to improve the local explanation for black box. The mask predictor can transform the output of the mask generator to establish the right-skewed distributions for relevance scores with the designed controller, which can decrease the difficulty of tuning the hyper-parameters. To solve the problem of the ad hoc constraints, the distribution controller models these above constraints and then is integrated with a neural network to directly guide the distribution of relevance scores.
(2) the paper develops two metrics in terms of faithfulness and explainability to evaluate the effectiveness of the proposed method quantitatively. And compared with other baseline methods, it also generates saliency maps to visually demonstrate the explainability. All the experimental results demonstrate that the performance of the proposed method is better than others.
(3) the paper states the motivation and the proposed algorithm clearly. The code is given.
However, for the experiments, the following should be addressed.
(1) It would be better to evaluate the proposed method with other networks as black boxes.
(2) More detailed analysis should be added on the experimental results.
(3) It should compare the explainability with some of the latest baseline methods.
(4) It would be better to conduct the experiments on more datasets.

 Moreover, there are some minor comments:
(1) Please state more details about the innovation of the method.
(2)	Figure 1 needs to be polished, for example the font.

**Experience Assessment:**

I have published one or two papers in this area.

**Review Assessment: Checking Correctness Of Derivations And Theory:**

I assessed the sensibility of the derivations and theory.

**Review Assessment: Checking Correctness Of Experiments:**

I carefully checked the experiments.

**Review Assessment: Thoroughness In Paper Reading:**

I read the paper thoroughly.

---

> ### Author Response · Authors · 2019-11-15
> **Response to Reviewer #1**
>
> We are truly grateful for your valuable comments and suggestions. We would like to answer the reviewer’s questions as follows.
> ===========================
> Q1. It would be better to evaluate the proposed method with other networks as black boxes.
>
> A. Thanks for your suggestions. Besides ResNet50, we introduce VGG19 and GoogleNet as our black boxes in Sec. 4.2 in the revision. We perform all the quantitative evaluations for the explanation methods on these networks and display the examples corresponding to different black boxes (Owing to limited space, most examples are left to Appendix). We hope it can show the broad applicability of our method.
> ===========================
> Q2. More detailed analysis should be added on the experimental results.
>
> A. Thanks for pointing it out. In the revision, we have added more targeted analysis of the experimental results based on the motivations of this paper. Besides, we also introduce extra experiments to demonstrate the effectiveness of the proposed innovations. These contents are highlighted in the color of blue. Please refer to Sec. 4.3 and 4.4 for more details.
> ===========================
> Q3. It should compare the explainability with some of the latest baseline methods.
>
> A. Thanks for pointing it out. In the revision, we have added the compared method [A], [B] as the baselines. Both are perturbation-based methods that optimize the mask directly for each image. Differently, the former introduces the intermediate feature maps to regularize the masks, while the latter proposes to use the extreme perturbation with an area constraint to optimize the masks. We follow [A] and reproduce the codes for our experiments. For [B], we use their publicly available codes to produce masks.
> [A] Du, M., Liu, N., Song, Q., & Hu, X. Towards explanation of dnn-based prediction with guided feature inversion. SIGKDD 2018.
> [B] Fong, Ruth, Mandela Patrick, and Andrea Vedaldi. Understanding Deep Networks via Extremal Perturbations and Smooth Masks. ICCV. 2019.
> ===========================
> Q4. It would be better to conduct the experiments on more datasets.
>
> A. Thank you very much for your valuable suggestions. To demonstrate the effectiveness of the proposed innovations, we will consider introducing more datasets, as well as more kinds of application tasks as our future work.
> ===========================
> Q5. Please state more details about the innovation of the method.
>
> A. Thanks for pointing it out. In the revision, we have clarified our innovations in Sec. 1, which can also be found in the general response. Moreover, to better introduce these innovations, we reorganize Sec. 3.2 according to these innovations.
> ===========================
> Q6. Fig. 1 needs to be polished, for example the font.
>
> A. Thanks for pointing it out. We have checked the font and size of this figure. Apart from the notations of the mathematical typos and four modules (encoder of the classifier, mask generator, distribution controller, and loss for classification), we consistently apply the same font and size. For the notations of all typos and four modules, we also fix their sizes, respectively. Moreover, thanks for your feedback, and we have checked all the figures to make them consistent.
> ===========================

---

### Official Review · AnonReviewer2 · 2019-10-24
**Official Blind Review #2**

**Rating:** 3

**Review:**

The paper introduced a new framework in the family of local perturbation-based explanations (saliency maps) for deep neural networks. The method is similar to existing methods in the literature (real-time saliency maps, etc) and seeks to tackle the difficulty of hyper-parameter selection for the complex optimization objective of these methods through assuming distributional preferences over the scores in generated saliency maps. The idea is that basically many of the terms in those objectives could be achieved by enforcing a right-skewed distribution on the importance scores (and also using the interpolation idea).

I lean towards rejecting this paper for one main reason: the contributions are not enough for this venue. Both the introduced method and the metrics are slight modifications of what already exists and the experimental results do not convince me that the introduced method tackles an important problem with the existing methods.

More specifically, the paper seeks to solve the hyper-parameter selection problem with existing approaches (e.g. MGNet). None of the experiments seek to show how the method is helping with this problem. For instance, the average performance of MGNet over a set of randomly selected hyperparameters versus the model's performance, the computational cost of training a good MGNet mask generator versus the introduced method, etc.

One major motivation behind perturbation-based methods is their black-box nature. The authors refer to their method as being applied to black-box models. The introduced method clearly utilizes the inner layer activations of the network. This is not a black-box method.

The main contribution is a marginally improved performance compared to rival methods for the introduced metrics (which are slight modifications to already existing metrics.) The introduced Explainability metric is very well-justified and tackles the issues with the two older metrics. The advantage of the introduced methods is not very clear on the provided subjective examples.

A few suggestions and questions:

* The exact contribution of the work should be stated more clearly and experiments should be targeted towards it.
* The paper is sometimes very, very difficult to grasp (Sec 3.2 and 3.3.)
* It might be a wrong intuition but it seems like the fine-tuning step seems to make the output scores curated for the M_F metric which would make the results in Table 1 not quite fair. The same modification seems to be applicable to any given importance mask and a more fair comparison would include applying the same fine-tuning step in all of the methods.
* The method's optimization process seems to be only focused on the SSR and neglect SDR. The text does not make it clear why this does not result in any loss of performance and if that's the case why it was necessary for methods like MGNet.

**Experience Assessment:**

I have published in this field for several years.

**Review Assessment: Checking Correctness Of Derivations And Theory:**

I assessed the sensibility of the derivations and theory.

**Review Assessment: Checking Correctness Of Experiments:**

I assessed the sensibility of the experiments.

**Review Assessment: Thoroughness In Paper Reading:**

I read the paper thoroughly.

---

> ### Author Response · Authors · 2019-11-15
> **Response to Reviewer #2 (2/2)**
>
>
> ===========================
> Q6. One major motivation behind perturbation-based methods is their black-box nature. The authors refer to their method as being applied to black-box models. The introduced method clearly utilizes the inner layer activations of the network. This is not a black-box method.
>
> A. Thanks for pointing it out. We agree with the reviewer that black-box explanation methods should only utilize the input and output of DNNs. However, in our paper, we introduce the black box to describe DNNs that lack meaningful explanations, rather than the proposed method itself. Thus, this paper follows [A] and uses the word ‘black-box‘. However, to address this issue and reduce the ambiguity, we have checked the paper carefully to avoid using the abbreviation ‘black-box’ for ‘black-box classifiers’. We hope it can address your concerns in a proper way. Nevertheless, if the reviewer believes the misunderstanding still exists, we would remove the word ‘black-box’ from the paper and consider other descriptions.
> [A] Dabkowski, P., & Gal, Y. Real-time image saliency for black-box classifiers. In Advances in Neural Information Processing Systems (pp. 6967-6976), 2017
> ===========================
> Q7. It might be a wrong intuition but it seems like the fine-tuning step seems to make the output scores curated for the M_F metric which would make the results in Table 1 not quite fair.
>
> A. Thanks for pointing it out. We apologize for the misunderstanding in this part, where we do not use any tuning step on any method for all quantitative metrics. To avoid this misunderstanding, we add two sentences for clarifying this part.
> Firstly, in the paragraph of the tuning step in Sec. 3.2, we emphasize that the proposed tuning step does not affect the ranking of pixels, meaning it brings no change to these metrics. Second, to further avoid this issue, we also clarify it during experiments in Sec. 4.2 that we do not use any tuning step on any method.
> ===========================
> Q8. The method's optimization process seems to be only focused on the SSR and neglect SDR. The text does not make it clear why this does not result in any loss of performance and if that's the case why it was necessary for methods like MGNet.
>
> A. Thanks for pointing it out. To address this concern, we add ablation studies on both MGNet and our methods to analyze the effect of different terms, including SSR and SDR. In brief, the evidence shows that although neglecting SDR in our method also reduces its quality to some degree, it is acceptable since the hyper-parameter selection problem is avoided. However, for MGnet, neglecting SDR will affect the results remarkably and make the tuning of its hyper-parameter more difficult. Please refer to Sec. 4.4 for detailed analysis.
> ===========================

---

> ### Author Response · Authors · 2019-11-15
> **Response to Reviewer #2 (1/2)**
>
> Thank you for your detailed and insightful review. We have updated the paper and address the specific questions below.
> ===========================
> Q1. The contributions are not enough for this venue. Both the introduced method and the metrics are slight modifications of what already exists. The exact contribution of the work should be stated more clearly.
>
> A. Thanks for pointing it out and your valuable suggestion. In the revision, we have re-stated our motivations and innovations more clearly. Specifically, our method (1) introduces the distribution controller to guide the scores towards desired distributions for discriminative scores and (2) optimizes the model with classification losses. It therefore enables discriminative scores on the supporting features for better explaining the decisions and avoids the ad hoc constraints for the non-trivial hyper-parameter selection. Please refer to the general response or Sec. 1 in the revision for more details.
> ===========================
> Q2. The paper seeks to solve the hyper-parameter selection problem with existing approaches (e.g. MGNet). None of the experiments seek to show how the method is helping with this problem; For instance, the average performance of MGNet over a set of randomly selected hyperparameters versus the model's performance, the computational cost of training a good MGNet mask generator versus the introduced method, etc.
>
> A. Thanks for your insightful suggestions. We have added a targeted ablation study on both MGNet and our method to demonstrate the effectiveness of solving the hyper-parameter selection problem. Specifically, we investigate the experimental results of both methods under different settings of ad hoc constraints. It is evidenced that MGNet is very sensitive to its hyper-parameters, which results in the difficulty of the hyper-parameter selection problem. On the contrary, our method is insensitive to the hyper-parameters, and we can optimize the predictor without any ad hoc constraints and obtain satisfying performance. Please refer to Sec. 4.4 for more details.
> Besides, we apologize for the misunderstanding of the benefit of costs. In Sec. 3.2, we aimed to show that by neglecting SDR, we could save half memory costs for each image when performing optimization. It in turn allowed a larger batch size. However, since we use the same batch size in experiments, we remove this point from the contribution to avoid this issue.
> ===========================
> Q3. The paper is sometimes very, very difficult to grasp (Sec. 3.2 and 3.3.)
>
> A. We agree with the reviewer that the content in Sec. 3.2 and 3.3 is sometimes difficult to grasp. In the revision, we have re-organized the content of this part from two aspects. 1. The design and implementation of distribution controllers. 2. The optimization with classification losses. To make this section clearer, we left a detailed discussion of the hyper-parameters of controllers to the Appendix. Since the key point behind this discussion is very intuitive, namely, using the intuitive geometric properties of their probability density functions to choose a proper setting, we believe it has improved this section considerably without affecting the understanding of the overall paper.
> ===========================
> Q4. Experiments should be targeted towards the contributions.
>
> A. Thanks for your suggestions. Targeting at two innovations mentioned in Q1, we have added two parts of experiments at Sec. 4.4 to demonstrate their effectiveness, respectively. Please refer to the revision for details.
> ===========================
> Q5. The experimental results do not convince me that the introduced method tackles an important problem with the existing methods. The advantage of the introduced methods is not very clear on the provided subjective examples.
>
> A. Thanks for pointing it out. In the revision, we have added more analysis in Sec. 4.2 and Sec. 4.3 to show the benefits of the proposed method. Besides, based on additional experiments in Sec. 4.4, we provide more evidence and discussions to clarify our contributions. Please refer to these sections in the revision for details. Thanks.
> ===========================

---

### Official Review · AnonReviewer4 · 2019-10-28
**Official Blind Review #4**

**Rating:** 3

**Review:**

Paper Summary:

This paper proposed a method to produce instance-wise saliency map for image classification tasks. The proposed method develop a U-net-based generator for saliency mask, where the important modifications are (1) a skewness-inducing activation function for mask generation (i.e. controller), which is either a ReLU function or a scaled sigmoid function raised to a certain power. Authors argue that the proposed activation function leads to output saliency score to have right-skew distribution, which leads to more distinguished saliency map. (2) a smoothness-inducing mechanism where the saliency mask is generated at coarser scale then up-sampled with a bilinear operation so the generated map is smooth. Finally, the training is performed by minimizing solely on the cross-entropy loss with respect to the original model predictions, which is expected to improve faithfulness in mimicking the original black-box classifiers.

I'm leaning toward rejecting this paper in its current form. While I think this paper proposed an interesting strategy in improving the faithfulness of the explainer (i.e. training on the cross-entropy loss with respect to original classifier), the rest of the two modification either already exists in the literature (generate mapping at coarser resolution is an idea from Dabkowski & Gal (2017) / Du (2018) as pointed out by the author) or suffers potential technical issues that can benefit from further methodology improvement/empirical justification (please see Major Comments). Furthermore, empirical experiments seem to suggest that the proposed method sometimes fail to produce more smooth and more discriminative maps compared to its MGnet baseline (e.g. Figure 5, top right row), calling into the question of whether training solely on a cross-entropy loss is enough to ensure the generated saliency map is of high-quality. I still like the overall idea of this paper, and I think this work can be made more rigorous and informative by including a careful ablation study about the loss/gain of different regularization terms and loss functions (e.g. the effect of TV(M) term on smoothness, the effect of f(\phi(I, 1-M)) term on discriminativeness, and the trade-off between cross-entropy v.s. the original negative log likelihood loss as used in Dabkowski & Gal (2017)).


Major Comments:

(1) Custom Transformer

The derivation of the new activation function assumes that the distribution of the hidden-layer output $m^g_{ij}$ is strictly Gaussian (i.e. unimodal, symmetric, and with sufficiently light tail), and author proposed to build this Gaussian distribution from raw hidden-unit output through standardization (Section 3.2.2). However, this approach may not be valid in practice. This is because there is no guarantee that the distribution of $m^g_{ij}$ is strictly unimodal and symmetric. If the output is skewed/multimodal in anyway, the normalization operation won't produce a Gaussian distribution, since a linear transformation (i.e. normalization) of a skewed distribution is still a skewed distribution. As a result, I am worried that the hyperparameter derived in Section 3.2.3 may not always lead to right-skewed distribution as expected. If author would like to ensure the outcome distribution is right-skewed, it may be good to tune these activation function parameters for each instance so the empirical distribution of $m^d_{ij}$ satisfy certain skewness criteria (e.g., maybe tune the parameter in a way such that the pearson's coefficient of skewness of the empirical distribution is sufficiently small).

In addition, I find the author's claim about the benefit of right-skewed distribution (i.e. producing more discriminative maps) not yet well supported by the empirical evidence. For example, in Figure 4 the produced maps in the first three rows are relatively fuzzy, and the difference between the irrelevant features and the important features does not seem to be very high. The situation is only improved after the instance-wise finetuning. I think there are two possible reasons, either (a) the right-skewness of the actual distribution of the saliency map score in those images were not guaranteed (as mentioned earlier, I believe the current approach cannot guarantee the pre-activation units are Gaussian distributed, therefore the actual skewness of the post-activation values may vary depending on the input). (b) the right-skewness in the outcome distribution alone is not enough to guarantee good discrimininative behavior.
To clarify this (and provide more evidence for author's claim that skewness <-> better discrimination), author can consider visualizing the saliency map with varying degree of skewness (you can measure the skewness of the empirical distribution of the saliency-map score using pearson's coefficient of skewness), and show that the "discriminativeness" and skewness of the empirical distribution are correlated.

(2) Smoothness of the produced map

In the original MGnet work, the authors used two mechanisms to guarantee smoothness of the saliency map: (1) generate map at coarser level and (2) TV penalty on loss function. In this work author used only (1). As a result, the produced map in some cases (e.g. Figure 5, top right row for BF After, also Figure 7, left fifth row for BF after) appear jagged and not smooth. I'm wondering if adding TV penalty back would help somewhat. Again, a careful ablation study would be beneficial in clarifying this.

Minor Comments

(1) There's some minor technical issue in the description for Equation (6). Assuming $z_{ij}$ is Gaussian, it is $\eta$ AND sigmoid() COMBINED to "approach the uniform distribution", it is important to mention the sigmoid function since the uniform transformation used here is based on the Probability Integral Transform,   (i.e. if a random variable $z \sim \Phi(.)$ follows Gaussian distribution with CDF $\Phi(.)$, then $\Phi(z) \sim Unif(0, 1)$, i.e. you are transforming a Gaussian random variable using Gaussian CDF to get uniform distribution), and equation (6) is using $\eta$ AND sigmoid() combined together to approximate this Gaussian CDF [1].

(2) Equation (14) on page 13. There should be an $z^2$ term within the exponential function.

Reference:
[1] Gary R.Waissi, Donald F.Rossin (1996) A sigmoid approximation of the standard normal integral.

**Experience Assessment:**

I have read many papers in this area.

**Review Assessment: Checking Correctness Of Derivations And Theory:**

I carefully checked the derivations and theory.

**Review Assessment: Checking Correctness Of Experiments:**

I assessed the sensibility of the experiments.

**Review Assessment: Thoroughness In Paper Reading:**

I read the paper thoroughly.

---

> ### Author Response · Authors · 2019-11-15
> **Response to Reviewer #4 (2/2)**
>
>
> ===========================
> Q4. Empirical experiments seem to suggest that the proposed method sometimes fail to produce more smooth and more discriminative maps compared to its MGnet baseline (e.g. Fig. 5, top right row). Besides, the benefit of right-skewed distribution (i.e. producing more discriminative maps) not yet well supported by the empirical evidence. in Fig. 4 the produced maps in the first three rows are relatively fuzzy, and the difference between the irrelevant features and the important features does not seem to be very high. I think there are two possible reasons, either (a) the right-skewness of the actual distribution of the saliency map score in those images were not guaranteed (as mentioned earlier, I believe the current approach cannot guarantee the pre-activation units are Gaussian distributed, therefore the actual skewness of the post-activation values may vary depending on the input). (b) the right-skewness in the outcome distribution alone is not enough to guarantee good discriminative behavior.
>
> A. Thanks for pointing it out. We totally agree with the review that compared with MGnet, the difference of our saliency maps between the irrelevant features and the supporting (important) features sometimes is reduced. Nevertheless, our motivation is to obtain more discriminative scores on all supporting features while ensuring they are still higher than irrelevant features. For this, the distribution-guided methods indeed obtain such property in most cases. On the contrary, MGnet produces similar scores on all supporting features, reducing the explainability of explaining black-box classifiers. Furthermore, we can take advantage of the improved ranking of scores (demonstrated by the quantitative metrics) to tune the saliency maps.
> However, we agree with the reviewer that the possible reason behind the above cases is that the right-skewness of the actual distribution of the saliency map scores in those images is not guaranteed. It is also evidenced by the observation that the saliency maps of basic transformers with only instance normalization and ReLU are generally worse than those of custom transformers. As we know, compared with custom transformers, basic transformers are not able to handle the general distributions, such as left-skewed inputs. Meanwhile, for the relationship between the right-skewness and discrimination, please refer to Q2 for detailed discussions.
> ===========================
> Q5. Generate mapping at coarser resolution is an idea from Dabkowski & Gal (2017) / Du (2018) as pointed out by the author).
>
> A. Thanks for pointing it out. We agree with the reviewer that since the discrimination and smoothness of saliency maps are emphasized at the same time, it makes the solution of smoothness sometimes seem equally important to that of discrimination. To avoid this issue, the revision states more clearly about the innovations of this paper. Besides, to focus on the discrimination of masks, we simplify the content of handling smoothness.
> ===========================
> Q6. There's some minor technical issue in the description for Equation (6).
>
> A. Thanks for your insightful suggestion. In the revision, we have added the expression of the sigmoid function. To provide a clear description of this section, we left all the detailed discussion of the parameter setting of controllers to the Appendix, where the sigmoid approximation of the standard normal integral is clarified.
> ===========================
> Q7. Equation (14) on page 13. There should be a z^2 term within the exponential function.
>
> A. Thanks for pointing it out. We have fixed this issue in the revision.
> ===========================

---

> ### Author Response · Authors · 2019-11-15
> **Response to Reviewer #4 (1/2)**
>
> Thank you for detailed comments and insightful feedback on our paper. In the following, we reply to your comments.
> ===========================
> Q1. The derivation of the new activation function assumes that the distribution of the hidden-layer output is strictly Gaussian. As a result, I am worried that the hyperparameter derived in Sec. 3.2.3 may not always lead to right-skewed distribution as expected.
>
> A. Thanks for pointing it out. We absolutely agree with the reviewer that our proposed custom transformer is built upon the Gaussian assumption on inputs and the instance normalization only obtains the normal distribution in this case. To address your concerns, we reorganize this section with two steps. Firstly, we assume that the input scores of controllers are strictly Gaussian to derive the new activation function. The motivation is that, a sum of even two independent random variables is more Gaussian than the original variables [A]. Then, we empirically demonstrate that the proposed controller built upon this assumption can also benefit other distributions towards the right-skewed distributions, by varying inputs with different distributions as you mentioned. The details are presented in Sec. 3.2 in the revision.
> [A] Hyvärinen, Aapo, and Erkki Oja. Independent component analysis: algorithms and applications. Neural networks vol. 13, no. 4-5, pp. 411-430, 2000.
> ===========================
> Q2. If author would like to ensure the outcome distribution is right-skewed, it may be good to tune these activation function parameters for each instance so the empirical distribution of satisfy certain skewness criteria (e.g., maybe tune the parameter in a way such that the Pearson's coefficient of skewness of the empirical distribution is sufficiently small). In addition, I find the author's claim about the benefit of right-skewed distribution (i.e. producing more discriminative maps) not yet well supported by the empirical evidence. To clarify this (and provide more evidence for the author's claim that skewness <-> better discrimination), the author can consider visualizing the saliency map with varying degrees of skewness.
>
> A. We sincerely appreciate your suggestion of using Pearson’s coefficient of skewness to evaluate the skewness. Nevertheless, when we estimate the Pearson’s coefficient (with median and mean), it fails frequently on nearly multimodal distributions and some other cases. However, we agree with the reviewer that the relationship between the preset distributions of controllers and the outcome distributions of scores should be checked.
> To provide the evidence that right-skew distribution can improve the discrimination, we have tried an alternative way by performing an extra experiment in Sec. 4.4. Specifically, we change the hyper-parameters in controllers to preset different distributions (with Gaussian assumption) and guide the input towards these distributions. For this, we train multiple mask predictors based on these controllers and display their predicted masks. The evidence shows that the preset distributions significantly affect the final outputs, and these controllers indeed guide the scores towards the expected distributions. Since Pearson’s coefficient of skewness sometimes fails, we directly display the preset distributions instead of the Pearson’s coefficient.
> ===========================
> Q3. This work can be made more rigorous and informative by including a careful ablation study about the loss/gain of different regularization terms and loss functions (e.g. the trade-off between cross-entropy v.s. the original negative log-likelihood loss). Besides, I'm wondering if adding TV penalty back would help somewhat. Again, a careful ablation study would be beneficial in clarifying this.
>
> A. We appreciate the reviewer’s suggestion for an ablation study, in order to analyzing how ad hoc constraints affect the produced maps and the tradeoff between the performance and the non-trivial hyper-parameter tuning. To this end, we have added the ablation study on proposed methods, where we add all ad hoc constraints back and remove them generally. Besides, we also perform the ablation study on MGnet as a comparison for further analysis. We would like to mention that the trade-off between cross-entropy v.s. the original negative log-likelihood loss implies the changing of $\lambda_{bg}$, as the cross-entropy itself is equal to the original positive log-likelihood loss. In general, we observe that for both methods, with carefully-tuned hyper-parameters on all ad hoc constraints, the performance is improved. However, our methods are insensitive to the hyper-parameters, and we can perform optimization with classification losses without any constraint and obtain satisfying masks. On the contrary, MGNet is very sensitive to its hyper-parameters, which results in the difficulty of the hyper-parameter selection problem. More detailed analysis can be found in Sec. 4.4.
> ===========================

---

### Author Response · Authors · 2019-11-15
**The General Response to All Reviewers**

We appreciate all reviewers for providing insightful comments on the technical aspects of the proposed method. Below, we summarize the revisions. Detailed responses to each reviewer are followed based on your feedback.
===========================
# We modify the left part of Fig.1 as follows.
    -	We display an example for the comparison of preferred saliency maps to others in Fig.1(a).
    -	We display different desired distributions for controllers and show the saliency maps obtained from the corresponding models in Fig.1(b).
# We add an example to better clarify the motivation in Sec. 1.
    -	Besides highlighting the pixels within the dog for the images of dogs, the relevance scores should also be discriminative at various parts of the dog, such as higher scores on the head than legs. The examples are shown in Fig.1(a).
# We reorganize Sec. 3.2 according to two innovations.
    -	We introduce distribution controllers for guiding scores towards desired distributions. It is derived from the inputs with Gaussian distributions, and then generalized to other distributions.
    -	We introduce the model optimization with classification losses without ad hoc constraints.
    -	Besides, we leave the detailed discussion of setting the involved hyper-parameters of controllers to the Appendix, which is based on intuitive geometric properties of the probability density functions. It thus provides a clearer description of Sec. 3.2.
# We add more detailed analysis for experimental results.
    -	The second paragraph of Sec. 4.2 on quantitative metrics.
    -	The second and fourth paragraphs of Sec. 4.3 on visual examples.
# We add additional experiments in Sec. 4.2 to show the broad applicability.
    -	We add two recently proposed methods for comparison.
    -	We provide the explanation for three types of CNNs.
# We conduct targeted experiments in Sec. 4.4 to demonstrate the contributions of two innovations.
    -	The first paragraph investigates the effect of different desired distributions on the saliency maps.
    -	The second and third paragraphs evaluate the effectiveness of optimization with classification losses.
===========================
# Clarification on Motivation and Novelty:
    -	The current perturbation-based methods have two shortcomings: (1) These methods either distinguish supporting features (foreground) from others (background) and ignore the discrimination on these supporting features or only focus on discriminative scores on a small part of supporting features. (2) These methods face the no-trivial hyper-parameters tuning caused by the large information gap between the classification loss and the ad hoc constraints.
    -	This paper addresses the above issues based on two innovations: (1) It introduces the controllers with right-skewed distributions in mask predictors, which guides the scores towards the desired distributions for discriminative scores on supporting features. (2) It introduces the classification loss to optimize the above mask predictor to avoid ad hoc constraints, which also improves the faithfulness.
    -	The evidence from the targeted experiments demonstrates the effectiveness of the above innovations on addressing the shortcomings. The experiments with the quantitative metrics in terms of faithfulness and explainability also demonstrate the advantages of our method.
===========================

---

### Decision · Program_Chairs · 2019-12-19

**Decision:**

Reject

**Comment:**

This paper proposed a method to estimate the instance-wise saliency map for image classification, for the purpose of improving the faithfulness of the explainer. Based on the U-net, two modifications are proposed in this work. While reviewer #3 is overall positive about this work, both Reviewer #1 and #2 rated weak reject and raised a number of concerns. The major concerns include the modifications either already exist or suffer potential issue. Reviewer #2 considered that the contributions are not enough for ICLR, and the performance improvement is marginal. The authors provided detailed responses to the reviewers’ concerns, which help to make the paper stronger, but did not change the rating. Given the concerns raised by the reviewers, the ACs agree that this paper can not be accepted at its current state.